# Revisiting and Advancing Fast Adversarial Training Through the lens of Bi-Level Optimization

## Abstract

Adversarial training (AT) has become a widely recognized defense mechanism to improve the robustness of deep neural networks against adversarial attacks. It originated from solving a min-max optimization problem, where the minimizer (i.e., defender) seeks a robust model to minimize the worst-case training loss in the presence of adversarial examples crafted by the maximizer (i.e., attacker). However, the min-max nature makes AT computationally intensive and thus difficult to scale. Thus, the problem of Fast-AT arises. Nearly all the recent progress is achieved based on the following simplification: The iterative attack generation method used in the maximization step of AT is replaced by the simplest one-shot gradient sign-based PGD method. Nevertheless, Fast-AT is far from satisfactory, and it lacks theoretically-grounded design. For example, a Fast-AT method may suffer from *robustness catastrophic overfitting* when training with strong adversaries.

In this paper, we foster a technological breakthrough for designing Fast-AT through the lens of *bi-level optimization* (BLO) instead of min-max optimization. First, we theoretically show that the most commonly-used algorithmic specification of Fast-AT is equivalent to the linearized BLO along the direction given by the sign of input gradient. Second, with the aid of BLO, we develop a new systematic and effective fast bi-level AT framework, termed Fast-BAT, whose algorithm is rigorously derived by leveraging the theory of implicit gradient. In contrast to Fast-AT, Fast-BAT has the least restriction to placing the tradeoff between computation efficiency and adversarial robustness. For example, it is capable of defending sign-based projected gradient descent (PGD) attacks without calling any gradient sign method and explicit robust regularization during training. Furthermore, we empirically show that our method outperforms state-of-the-art Fast-AT baselines. In particular, Fast-BAT can achieve superior model robustness without inducing robustness catastrophic overfitting and losing standard accuracy.

## 1 Introduction

Given the fact that machine learning (ML) models can be easily fooled by tiny adversarial perturbations (also known as adversarial attacks) on the input (Goodfellow et al., 2014; Carlini & Wagner, 2017; Papernot et al., 2016), learning robust deep neural networks (DNNs) is now a major focus in research. Nearly all existing effective defense mechanisms (Madry et al., 2018; Zhang et al., 2019b; Shafahi et al., 2019; Wong et al., 2020; Zhang et al., 2019a; Athalye et al., 2018a) are built on the adversarial training (AT) recipe, first developed in (Szegedy et al., 2014) and later formalized in (Madry et al., 2018) using min-max optimization. In contrast to standard model training using empirical risk minimization, AT (Madry et al., 2018) calls *min-max optimization*. That is, a minimizer (*i.e.* defender) seeks to update model parameters against a maximizer (*i.e.* attacker) that aims to worsen the training loss by perturbing each training example.

The AT-type defenses have been widely adopted in various application domains including image classification (Goodfellow et al., 2014; Madry et al., 2018; Kurakin et al., 2017), object detection (Zhang & Wang, 2019), natural language processing (Miyato et al., 2016; Zhu et al., 2019), and healthcare (Finlayson et al., 2019; Mahmood et al., 2019). Despite their effectiveness, the min-max optimization nature makes them difficult to scale. This is because *multiple* maximization steps

(required by an iterative attack generator) are needed at every model training step in AT. The resulting prohibitive computation cost prevents AT from a feasible solution to enhance adversarial robustness when computing resource is limited. For example, Xie et al. (2019) used 128 GPUs to make AT practical on ImageNet. Thereby, how to speed up AT without losing accuracy and robustness is now a *grand challenge* for adversarial defense.

Very recently, some work attempted to develop computationally-efficient alternatives of AT, which we call 'fast' versions of AT (Shafahi et al., 2019; Zhang et al., 2019a; Wong et al., 2020; Andriushchenko & Flammarion, 2020). To the best of our knowledge, FAST-AT (Wong et al., 2020) and FAST-AT with gradient alignment (GA) regularization, termed FAST-AT-GA (Andriushchenko & Flammarion, 2020), are the two state-of-the-art (SOTA) 'fast' versions of AT, since they achieve a significant reduction in computation complexity and preserve accuracy and robustness to some extent. To be specific, FAST-AT (Wong et al., 2020) replaces an iterative attack generator used in AT with a heuristics-based single-step attack generation method. Thus, it merely takes computation cost comparable to standard model training. However, FAST-AT suffers two main issues: *(i)* lack of stability, *i.e.*, large variance in performance (Li et al., 2020), and *(ii)* robustness catastrophic overfitting, *i.e.*, a large drop of robustness when training with strong adversaries (Andriushchenko & Flammarion, 2020). To alleviate these problems, Andriushchenko & Flammarion (2020) proposed FAST-AT-GA by penalizing FAST-AT using an explicit robust regularization given by GA. However, we will show that FAST-AT-GA encounters a new problem *(iii)*: FAST-AT-GA hampers standard accuracy, making a poor accuracy-robustness tradeoff at large attack budget ($\epsilon = 16/255$), *i.e.* the improvement on RA is at cost of a sharp drop on SA. Given the limitations *(i)-(iii)*, we ask:

How to design a *theoretically-grounded* 'fast' version of AT with *improved stability*, *mitigated catastrophic overfitting*, and *enhanced accuracy-robustness tradeoff*?

To address above question, paper we revisit and advance AT through the lens of bi-level optimization (BLO) (Dempe, 2002), where we cast the attack generation problem as a *lower-level* optimization problem with constraints and the defense as an *upper-level* optimization problem in the objective. To the best of our knowledge, this is the first work to make a solid connection between adversarial defense and BLO. Technically, we show that FAST-AT can be *interpreted* as BLO with linearized lower-level problems. Delving into linearization of BLO, we propose a novel, theoretically-grounded 'fast' AT framework, fast bi-level AT (FAST-BAT). Practically, Table 1 highlights some achieved improvements over FAST-AT and FAST-AT-GA: When a stronger train-time attack (*i.e.*, $\epsilon = 16/255$ vs. $8/255$) is adopted, FAST-AT suffers a large degradation of robust accuracy (RA) and standard accuracy (SA), together with higher variances than proposed FAST-BAT. Although FAST-AT-GA outperforms FAST-AT, it still incurs a significant SA loss (over 21%) at $\epsilon = 16/255$. By contrast, FAST-BAT yields a more graceful SA-RA tradeoff: 9% improvement of SA without loss of RA. Different from FAST-AT-GA, FAST-BAT achieves above improvements in stability, RA and SA without resorting to any extra robust regularization and thus takes less computation cost.

Table 1: Performance overview of proposed FAST-BAT vs. the baselines FAST-AT (Wong et al., 2020) and FAST-AT-GA (Andriushchenko & Flammarion, 2020) on (CIFAR-10, PreActResNet-18). All methods are robustly trained under two perturbation budgets $\epsilon = 8/255$ and $16/255$ over 20 epochs. We use the early-stop policy (Rice et al., 2020) to report the model of best robustness for each method. The evaluation metrics include robust accuracy (RA) against PGD-50-10 attacks (50-step PGD attack with 10 restarts) (Madry et al., 2018) at $\epsilon = 8/255$ and $16/255$ (test-time $\epsilon$ is *same* as the train-time), RA against AutoAttack (AA) (Croce & Hein, 2020) at $\epsilon = 8/255$ and $16/255$, and computation time (per epoch). The result $a_{\pm b}$ represents mean $a$ and standard deviation $b$ over 10 random trials. All experiments are run on a single Tesla-P100 GPU.

| Method | RA-PGD (%) ($\epsilon = 8/255$) | RA-PGD (%) ($\epsilon = 16/255$) | RA-AA (%) ($\epsilon = 8/255$) | RA-AA (%) ($\epsilon = 16/255$) | SA (%) ($\epsilon = 8/255$) | SA (%) ($\epsilon = 16/255$) | Time (s) |
|---|---|---|---|---|---|---|---|
| FAST-AT | 45.47±0.39 | 21.79±0.93 | 41.97±0.15 | 12.57±0.33 | 81.72±0.36 | 46.02±2.79 | 42 |
| FAST-AT-GA | 47.43±0.42 | 26.22±0.19 | 43.52±0.32 | 18.03±0.39 | 79.84±0.49 | 58.57±1.19 | 150 |
| **FAST-BAT** | 48.74±0.11 | 26.15±0.12 | 44.89±0.12 | 18.21±0.15 | 79.43±0.08 | 67.79±0.08 | 135 |

**Contributions.** We summarize our contributions below.

① We propose a new formulation of adversarially robust training through the lens of BLO, yielding a novel and theoretically-grounded interpretation of FAST-AT.

② We propose a new systematic and effective fast BLO-oriented AT framework, termed FAST-BAT, with rigorously-established theory and algorithm.

③ We conduct extensive experiments on FAST-BAT, showing its improved stability, mitigated catastrophic overfitting, and enhanced accuracy-robustness tradeoff; see illustrations in Table 1.

## 2 RELATED WORK

**Adversarial attack.** Adversarial attacks are techniques to generate malicious perturbations that are imperceptible to humans but can mislead the machine learning (ML) models (Goodfellow et al., 2014; Carlini & Wagner, 2017; Croce & Hein, 2020; Xu et al., 2019; Athalye et al., 2018b). A popular threat model that an adversary used is known as $\ell_p$-norm ball constrained attack ($p \in \{0, 1, 2, \infty\}$). This is also the focus of this paper. The adversarial attack has become a major approach to evaluate the robustness of deep neural networks (DNNs) and thus, help build safe artificial intelligence in many high stakes applications such as autonomous driving (Deng et al., 2020; Kumar et al., 2020), surveillance (Thys et al., 2019; Xu et al., 2020), and healthcare (Finlayson et al., 2019).

**Adversarial defense and robust training at scale.** Our work falls into the category of robust training, which was mostly built upon min-max optimization. For example, Madry et al. (2018) established the framework of AT for the first time, which has been recognized as one of the most powerful defenses (Athalye et al., 2018a). Extended from AT, TRADES (Zhang et al., 2019b) sought the optimal balance between robustness and generalization ability. Further, AT-type defense has been generalized to the semi-/self-supervised settings (Carmon et al., 2019; Chen et al., 2020) and integrated 1 with certified defense techniques such as randomized smoothing (Salman et al., 2019).

Despite the effectiveness of AT and its variants, they need to take high computation costs. How to speed up AT without losing performance remains an open question. Some recent works attempted to impose algorithmic simplifications to AT, leading to *fast but approximate* AT algorithms, such as 'free' AT (Shafahi et al., 2019), you only propagate once (YOPO) (Zhang et al., 2019a), FAST-AT (Wong et al., 2020), and FAST-AT regularized by gradient alignment (termed FAST-AT-GA) (Andriushchenko & Flammarion, 2020). In particular, FAST-AT and FAST-AT-GA are the baselines most relevant to ours since they were designed with the least computation complexity. However, their defense performance is far from satisfactory. For example, FAST-AT has poor training stability (Li et al., 2020) and suffers catastrophic overfitting when facing strong attacks (Andriushchenko & Flammarion, 2020). In contrast to FAST-AT, FAST-AT-GA yields improved robustness but has a poor accuracy-robustness tradeoff (e.g., Table 1). In this paper, we aim to advance the algorithm foundation of 'fast robust training' through the lens of BLO (bi-level optimization). We will show that the proposed FAST-BAT can lead to stable robust learning without suffering catastrophic overfitting and graceful tradeoff between accuracy and robustness.

**Bi-level optimization (BLO).** BLO is a unified hierarchical learning framework, where the objective and variables of an upper-level problem depend on the optimizer of certain lower-level problems. The BLO problem in its most generic form is a class of very challenging problems, and thus, the design of algorithms and theory for BLO focuses on special cases (Vicente et al., 1994; White & Anandalingam, 1993; Gould et al., 2016; Ghadimi & Wang, 2018; Ji et al., 2020; Hong et al., 2020). In practice, some successful applications of BLO to ML have been witnessed in meta-learning (Rajeswaran et al., 2019), data poisoning attack design (Huang et al., 2020), and reinforcement learning (Chen et al., 2019). However, as will be evident later, the existing BLO approach is not directly applied to adversarial defense due to the presence of the *constrained* nonconvex lower-level problem (for attack generation). To the best of our knowledge, our work makes a rigorous connection between adversarial defense and BLO for the first time.

## 3 A BI-LEVEL OPTIMIZATION VIEW ON FAST-AT

**Preliminaries on FAST-AT.** FAST-AT is designed for solving the adversarial training problem (Madry et al., 2018) given below

$$\underset{\boldsymbol{\theta}}{\text{minimize}} \; \mathbb{E}_{(\mathbf{x},y) \in \mathcal{D}} \left[ \underset{\boldsymbol{\delta} \in \mathcal{C}}{\text{maximize}} \; \ell_{\text{tr}}(\boldsymbol{\theta}, \mathbf{x} + \boldsymbol{\delta}, y) \right], \tag{1}$$

where $\boldsymbol{\theta} \in \mathbb{R}^n$ denotes model parameters, $\mathcal{D}$ is the training set consisting of labeled data pairs with feature $\mathbf{x}$ and label $y$, $\boldsymbol{\delta} \in \mathbb{R}^d$ represents adversarial perturbations subject to the perturbation

constraint $\mathcal{C}$, *e.g.*, $\mathcal{C} = \{\boldsymbol{\delta} \,|\, \|\boldsymbol{\delta}\|_\infty \leq \epsilon, \boldsymbol{\delta} \in [\mathbf{0}, \mathbf{1}]\}$ for $\epsilon$-toleration $\ell_\infty$-norm constrained attack (normalized to $[\mathbf{0}, \mathbf{1}]$), $(\mathbf{x} + \boldsymbol{\delta})$ is then called adversarial example, and $\ell_{\mathrm{tr}}(\cdot)$ represents a training loss.

The standard solver to problem (1) is known as AT (Madry et al., 2018). However, it has to call an *iterative* optimization method (*e.g.*, $K$-step PGD attack) to solve the inner maximization problem of (1). As a result, AT is computationally intensive. To improve its scalability, FAST-AT that only takes the *single-step* PGD attack for inner maximization was proposed and successfully implemented in (Wong et al., 2020). The algorithm backbone of FAST-AT is summarized below.

---

**FAST-AT algorithm**

Let $\boldsymbol{\theta}_t$ be the model parameters at iteration $t$. The $(t+1)$th iteration is given by
① (*Inner maximization by* 1-*step PGD*): $\boldsymbol{\delta} \leftarrow \mathcal{P}_\mathcal{C}\left(\boldsymbol{\delta}_0 + \alpha \cdot \mathrm{sign}\left(\nabla_{\boldsymbol{\delta}}\ell_{\mathrm{tr}}(\boldsymbol{\theta}_t, \mathbf{x} + \boldsymbol{\delta}_0, y)\right)\right)$,
where $\mathcal{P}_\mathcal{C}(\mathbf{a})$ denotes the projection operation that projects the point $\mathbf{a}$ onto $\mathcal{C}$, *i.e.*, $\mathcal{P}_\mathcal{C}(\mathbf{z}) = \arg\min_{\boldsymbol{\delta} \in \mathcal{C}} \|\boldsymbol{\delta} - \mathbf{z}\|_2^2$, $\boldsymbol{\delta}_0$ is a random uniform initialization within $[\mathbf{0}, \mathbf{1}]$, $\alpha > 0$ is a proper learning rate (*e.g.*, $1.25\epsilon$), and $\mathrm{sign}(\cdot)$ is the element-wise sign operation.
② (*Outer minimization for model training*): This can be conducted by any standard optimizer, *e.g.*, SGD. That is, $\boldsymbol{\theta}_{t+1} \leftarrow \boldsymbol{\theta}_t - \beta\nabla_{\boldsymbol{\theta}}\ell_{\mathrm{tr}}(\boldsymbol{\theta}_t, \mathbf{x} + \boldsymbol{\delta}, y)$, where $\beta > 0$ is a proper learning rate (*e.g.*, cyclic learning rate), and $\boldsymbol{\delta}$ is provided from the inner maximization step.

---

Roughly speaking, FAST-AT is a simplification of AT using 1-step PGD for inner maximization. However, as shown in (Wong et al., 2020), the successful implementation of FAST-AT is *different* from the 1-step PGD-based AT (Madry et al., 2018) due to the former's sophisticated hyperparameter choices in $\alpha$, $\boldsymbol{\delta}_0$, and $\beta$. Despite the efficacy of FAST-AT in some cases, Andriushchenko & Flammarion (2020) demonstrated that it could lead to the issue of robustness catastrophic overfitting when facing strong adversaries during training. In the literature, there was no grounded theory to justify the pros and cons of FAST-AT. We will show that BLO provides a promising solution.

**BLO: Towards a unified formulation of robust training.** BLO (bi-level optimization) is a unified hierarchical learning framework, involving two levels (*i.e.*, upper and lower levels) of optimization tasks, where one task is nested inside the other (*i.e.*, the objective and variables of an upper-level problem depend on the optimizer of the lower-level problem). The hierarchical learning framework provided by BLO can be used to precisely depict a robust training paradigm. Specifically, we can cast robustification as an *upper-level* problem whose optimization relies on a *lower-level* problem defined for attack generation. Thus, the BLO formulation of (1) is given by

$$\begin{aligned} \underset{\boldsymbol{\theta}}{\text{minimize}} \quad & \mathbb{E}_{(\mathbf{x},y)\in\mathcal{D}}[\ell_{\mathrm{tr}}(\boldsymbol{\theta}, \mathbf{x} + \boldsymbol{\delta}^*(\boldsymbol{\theta}; \mathbf{x}, y), y)] \\ \text{subject to} \quad & \boldsymbol{\delta}^*(\boldsymbol{\theta}; \mathbf{x}, y) = \arg\min_{\boldsymbol{\delta} \in \mathcal{C}} \ell_{\mathrm{atk}}(\boldsymbol{\theta}, \boldsymbol{\delta}; \mathbf{x}, y), \end{aligned} \quad (2)$$

where $\ell_{\mathrm{atk}}$ denotes an *attack objective*. For notation simplicity, we will use *data-omitted expressions* of $\ell_{\mathrm{tr}}$, $\ell_{\mathrm{atk}}$, and $\boldsymbol{\delta}^*$. The formulation (2) has two key differences from (1):

– First, the lower-level attack objective $\ell_{\mathrm{atk}}$ is customizable, not necessarily to be same as the opposite of the training objective, $-\ell_{\mathrm{tr}}$. As will be evident later, the flexibility of attack configuration in (2) enables us to interpret FAST-AT through the lens of BLO.

– Second, BLO calls an optimization routine different from min-max optimization used by (1). Even if we set $\ell_{\mathrm{atk}} = -\ell_{\mathrm{tr}}$, problem (2) does not reduce to (1) due to the presence of lower-level constraint $\boldsymbol{\delta} \in \mathcal{C}$ (see rigorous analysis in Appendix B). Specifically, solving the upper-level problem of (2) by gradient descent yields

$$\frac{d\ell_{\mathrm{tr}}(\boldsymbol{\theta}, \boldsymbol{\delta}^*(\boldsymbol{\theta}))}{d\boldsymbol{\theta}} = \nabla_{\boldsymbol{\theta}}\ell_{\mathrm{tr}}(\boldsymbol{\theta}, \boldsymbol{\delta}^*(\boldsymbol{\theta})) + \underbrace{\frac{d\boldsymbol{\delta}^*(\boldsymbol{\theta})^\top}{d\boldsymbol{\theta}}}_{\mathrm{IG}} \nabla_{\boldsymbol{\delta}}\ell_{\mathrm{tr}}(\boldsymbol{\theta}, \boldsymbol{\delta}^*(\boldsymbol{\theta})), \quad (3)$$

where the superscript $\top$ denotes the transpose operation, and $\nabla_{\boldsymbol{\theta}}\ell_{\mathrm{tr}}(\boldsymbol{\theta}, \boldsymbol{\delta}^*(\boldsymbol{\theta}))$ denotes the partial derivative with respect to (w.r.t.) the first input argument $\boldsymbol{\theta}$. In (3), $\frac{d\boldsymbol{\delta}^*(\boldsymbol{\theta})^\top}{d\boldsymbol{\theta}} \in \mathbb{R}^{n \times d}$ is referred to as *implicit gradient (IG)* because it is defined through an implicit constrained optimization problem $\min_{\boldsymbol{\delta} \in \mathcal{C}} \ell_{\mathrm{atk}}$. The dependence on IG is a 'fingerprint' of BLO (1) in contrast to AT or FAST-AT.

**BLO-enabled interpretation of FAST-AT.** In what follows, we demonstrate how FAST-AT relates to BLO. Our *main finding* is summarized below.



**Bi-level interpretation of FAST-AT**

FAST-AT can be interpreted as the **lower-level linearized BLO** with $\mathbf{z} = \boldsymbol{\delta}_0$ and $\lambda = 1/\alpha$:

$$\begin{aligned}
&\underset{\boldsymbol{\theta}}{\text{minimize}} && \mathbb{E}_{(\mathbf{x},y)\in\mathcal{D}}[\ell_{\mathrm{tr}}(\boldsymbol{\theta}, \boldsymbol{\delta}^*(\boldsymbol{\theta}))] \\
&\text{subject to} && \boldsymbol{\delta}^*(\boldsymbol{\theta}) = \underset{\boldsymbol{\delta}\in\mathcal{C}}{\arg\min}\ [(\boldsymbol{\delta} - \mathbf{z})^\top \mathrm{sign}(\nabla_{\boldsymbol{\delta}=\mathbf{z}}\ell_{\mathrm{atk}}(\boldsymbol{\theta}, \boldsymbol{\delta})) + (\lambda/2)\|\boldsymbol{\delta} - \mathbf{z}\|_2^2],
\end{aligned} \qquad (4)$$

where $\mathbf{z}$ is the linearization point, $\nabla_{\boldsymbol{\delta}=\mathbf{z}}\ell_{\mathrm{atk}}$ denotes the partial derivative of $\ell_{\mathrm{atk}}$ (w.r.t. $\boldsymbol{\delta}$) evaluated at $\mathbf{z}$, $\mathrm{sign}(\nabla_{\boldsymbol{\delta}=\mathbf{z}}\ell_{\mathrm{atk}}(\boldsymbol{\theta}, \boldsymbol{\delta}))$ is the linearization direction, and $\lambda > 0$ is a regularization parameter associated with the quadratic residual of linearization.



Our justification on the above claim is elaborated on below.

– First, the simplified lower-level problem of (4) leads to the **closed-form** solution

$$\begin{aligned}
\boldsymbol{\delta}^*(\boldsymbol{\theta}) &= \underset{\boldsymbol{\delta}\in\mathcal{C}}{\arg\min}\ (\lambda/2)\|\boldsymbol{\delta} - \mathbf{z} + (1/\lambda)\mathrm{sign}(\nabla_{\boldsymbol{\delta}=\mathbf{z}}\ell_{\mathrm{atk}}(\boldsymbol{\theta}, \boldsymbol{\delta}))\|_2^2 \\
&= \mathcal{P}_{\mathcal{C}}\left(\mathbf{z} - (1/\lambda)\mathrm{sign}(\nabla_{\boldsymbol{\delta}=\mathbf{z}}\ell_{\mathrm{atk}}(\boldsymbol{\theta}, \boldsymbol{\delta}))\right),
\end{aligned} \qquad (5)$$

which is given by the 1-step PGD attack with initialization $\mathbf{z}$ and learning rate $(1/\lambda)$. In the linearization used in (4), a quadratic regularization term (with regularization parameter $\lambda$) is introduced to ensure the strong convexity of the inner-level attack objective within the constraint set $\boldsymbol{\delta} \in \mathcal{C}$. Assisted by that, the lower-level solution is unique and its closed form is given by (5). Note that imposing such a strongly convex regularizer is also commonly used to stabilize the convergence of min-max optimization and BLO (Qian et al., 2019; Hong et al., 2020). If we set $\mathbf{z} = \boldsymbol{\delta}_0$ and $\lambda = 1/\alpha$, then (5) precisely depicts the inner maximization step used in FAST-AT .

– Second, by substituting (5) into the upper-level problem of (4), we can then follow (3) to update the model parameters $\boldsymbol{\theta}$. However, this calls the computation of IG $\frac{d\boldsymbol{\delta}^*(\boldsymbol{\theta})^\top}{d\boldsymbol{\theta}}$. If we regard $\mathcal{P}_{\mathcal{C}}$ is differentiable, then based on the closed-form of $\boldsymbol{\delta}^*(\boldsymbol{\theta})$ in (5), IG becomes

$$\frac{d\boldsymbol{\delta}^*(\boldsymbol{\theta})^\top}{d\boldsymbol{\theta}} = \mathbf{0}, \qquad (6)$$

where we use two facts: (1) The linearization point $\mathbf{z}$ is independent of $\boldsymbol{\theta}$, *i.e.* $\mathbf{z} = \boldsymbol{\delta}_0$; And (2) $\frac{d\mathrm{sign}(\cdot)}{d\boldsymbol{\theta}} = \mathbf{0}$ holds almost everywhere. Please refer to Appendix C for a rigorous proof of (6) using KKT conditions. Clearly, the use of gradient sign method simplifies the IG computation. Substituting (6) into (3), the upper-level optimization of (4) yields $\boldsymbol{\theta} \leftarrow \boldsymbol{\theta} - \beta\nabla_{\boldsymbol{\theta}}\ell_{\mathrm{tr}}(\boldsymbol{\theta}, \boldsymbol{\delta}^*(\boldsymbol{\theta}))$ (with learning rate $\beta$), which is precisely same as the outer minimization step used in FAST-AT.

The aforementioned analysis shows that the linearized BLO (4) is equivalent to FAST-AT by setting the linearization point $\mathbf{z}$ and the regularization parameter $\lambda$ as $\mathbf{z} = \boldsymbol{\delta}_0$ and $\lambda = 1/\alpha$.

## 4 FAST-BAT: ADVANCING FAST-AT BY BLO

**FAST-BAT and rationale.** The key take-away from (4) is that the conventional FAST-AT adopts the *sign of input gradient* to linearize the lower-level attack objective. However, a more natural and wiser choice is to use the first-order Taylor expansion for linearization. By doing so, problem (4) can be modified to the form of FAST-BAT

$$\begin{aligned}
&\underset{\boldsymbol{\theta}}{\text{minimize}} && \mathbb{E}_{(\mathbf{x},y)\in\mathcal{D}}[\ell_{\mathrm{tr}}(\boldsymbol{\theta}, \boldsymbol{\delta}^*(\boldsymbol{\theta}))] \\
&\text{subject to} && \boldsymbol{\delta}^*(\boldsymbol{\theta}) = \underset{\boldsymbol{\delta}\in\mathcal{C}}{\arg\min}\ [(\boldsymbol{\delta} - \mathbf{z})^\top \nabla_{\boldsymbol{\delta}=\mathbf{z}}\ell_{\mathrm{atk}}(\boldsymbol{\theta}, \boldsymbol{\delta}) + (\lambda/2)\|\boldsymbol{\delta} - \mathbf{z}\|_2^2],
\end{aligned} \qquad (7)$$

where similar to (5), the lower-level problem can be solved analytically as

$$\boldsymbol{\delta}^*(\boldsymbol{\theta}) = \mathcal{P}_{\mathcal{C}}\left(\mathbf{z} - (1/\lambda)\nabla_{\boldsymbol{\delta}=\mathbf{z}}\ell_{\mathrm{atk}}(\boldsymbol{\theta}, \boldsymbol{\delta})\right). \qquad (8)$$

In contrast to (6), the IG associated with (7) is no longer vacant since the gradient sign operation is not present in (8). To compute IG, the auto-differentiation (which calls the chain rule) can be applied to the closed-form of $\boldsymbol{\delta}^*(\boldsymbol{\theta})$. However, this will not give us an accurate and generalizable IG solution since the projection operation $\mathcal{P}_{\mathcal{C}}$ is *not* smooth and thus, the use of chain rule does not yield a rigorous derivation. Therefore, the *IG challenge* arises, which will be addressed in what follows.

**IG theory for FAST-BAT.** The problem of FAST-BAT (7) falls into a class of very challenging BLO problems, which require *constrained* lower-level optimization. The unconstrained case is easier to handle since one can apply the implicit function theory to the stationary condition of the lower-level problem to obtain IG (Hong et al., 2020). Yet, in the case of *constrained* problems, a stationary point could violate the constraints, and thus the stationary condition becomes non-applicable.

In problem (7), we are dealing with a special class of lower-level constraints – *linear constraints*:

$$\mathcal{C} = \{\|\boldsymbol{\delta}\|_\infty \le \epsilon, \boldsymbol{\delta} \in [-\mathbf{x}, \mathbf{1} - \mathbf{x}]\} \Longleftrightarrow \mathbf{B}\boldsymbol{\delta} \le \mathbf{b}, \text{ with } \mathbf{B} := \begin{bmatrix} \mathbf{I} \\ -\mathbf{I} \end{bmatrix}, \mathbf{b} := \begin{bmatrix} \min\{\epsilon\mathbf{1}, \mathbf{1} - \mathbf{x}\} \\ -\max\{-\epsilon\mathbf{1}, -\mathbf{x}\} \end{bmatrix}. \quad (9)$$

By exploiting above linearly constrained problem structure, we show that the IG challenge associated with (7) can be addressed via *Karush–Kuhn–Tucker (KKT)* conditions. We summarize our main theoretical result below and refer readers to Appendix A for detailed derivation.

**Theorem 1** *With a Hessian-free assumption, i.e.,* $\nabla_{\boldsymbol{\delta\delta}}\ell_{\mathrm{atk}} = \mathbf{0}$*, the IG (implicit gradient) of (7) is*

$$\frac{d\boldsymbol{\delta}^*(\boldsymbol{\theta})^\top}{d\boldsymbol{\theta}} = -(1/\lambda)\nabla_{\boldsymbol{\theta\delta}}\ell_{\mathrm{atk}}(\boldsymbol{\theta}, \boldsymbol{\delta}^*)\mathbf{H}_\mathcal{C}, \text{ with } \mathbf{H}_\mathcal{C} := \begin{bmatrix} \mathbf{1}_{p_1 < \delta_1^* < q_1}\mathbf{e}_1 & \cdots & \mathbf{1}_{p_1 < \delta_d^* < q_d}\mathbf{e}_d \end{bmatrix}, \quad (10)$$

*where* $\boldsymbol{\delta}^*$ *is given by (8), and* $\nabla_{\boldsymbol{\theta\delta}}\ell(\boldsymbol{\theta}, \boldsymbol{\delta}^*) \in \mathbb{R}^{n \times d}$ *denotes the second-order partial derivative evaluated at* $\boldsymbol{\theta}$ *and* $\boldsymbol{\delta}^*(\boldsymbol{\theta})$*, respectively. In* $\mathbf{H}_\mathcal{C} \in \mathbb{R}^{d \times d}$*,* $\mathbf{1}_{p_i < \delta_i^* < q_i} \in \{0, 1\}$ *denotes the indicator function over the constraint of* $\{\delta_i \mid p_i < \delta_i^* < q_i\}$*, which returns 1 if the constraint is satisfied,* $\delta_i^*$ *denotes the ith entry of* $\boldsymbol{\delta}^*(\boldsymbol{\theta})$*,* $p_i = \max\{-\epsilon, -x_i\}$ *and* $q_i = \min\{\epsilon, 1 - x_i\}$ *characterize the boundary of the linear constraint (9) for the variable* $\delta_i$*, and* $\mathbf{e}_i \in \mathbb{R}^d$ *denotes the basis vector with the ith entry being 1 and others being 0s.*

In Theorem 1, the rationale behind the Hessian-free assumption is that ReLU-based neural networks commonly lead to a piece-wise linear decision boundary w.r.t. the inputs (Moosavi-Dezfooli et al., 2019; Alfarra et al., 2020), and thus, its second-order derivative (Hessian) $\nabla_{\boldsymbol{\delta\delta}}\ell_{\mathrm{atk}}$ is close to zero. In Appendix E, we will empirically show that the Hessian-free assumption is reasonable for both ReLU and non-ReLU neural networks.

**FAST-BAT algorithm and implementation.** Similar to FAST-AT or AT, the FAST-BAT algorithm follows the principle of alternating optimization. Specifically, it consists of the IG-based upper-level gradient descent (3), interlaced with the lower-level optimal attack (8). We summarize the FAST-BAT algorithm below.

---

**FAST-BAT algorithm**

The $(t + 1)$th upper-level iteration of FAST-BAT is given below
① (*Lower-level solution*): Obtain $\boldsymbol{\delta}^*(\boldsymbol{\theta}_t)$ from (8);
② (*Upper-level model training*): Integrating the IG (10) into (3), call SGD to update

$$\boldsymbol{\theta}_{t+1} = \boldsymbol{\theta}_t - \alpha_1\nabla_{\boldsymbol{\theta}}\ell_{\mathrm{tr}}(\boldsymbol{\theta}_t, \boldsymbol{\delta}^*(\boldsymbol{\theta}_t)) - \alpha_2(-1/\lambda)\nabla_{\boldsymbol{\theta\delta}}\ell_{\mathrm{atk}}(\boldsymbol{\theta}_t, \boldsymbol{\delta}^*(\boldsymbol{\theta}_t))\mathbf{H}_\mathcal{C}\nabla_{\boldsymbol{\delta}}\ell_{\mathrm{tr}}(\boldsymbol{\theta}_t, \boldsymbol{\delta}^*(\boldsymbol{\theta}_t)), \quad (11)$$

where $\alpha_1, \alpha_2 > 0$ are learning rates associated with the standard model gradient and the IG-augmented descent direction, respectively.

---

It is clear from (11) that to train a robust model, FAST-BAT can be dissected into the regular FAST-AT update (*i.e.*, $\alpha_1$-associated term) and the additional update that involves IG, (*i.e.*, $\alpha_2$-associated term). To successfully implement FAST-BAT, we highlight some key hyper-parameter setups different from FAST-AT (Wong et al., 2020) and FAST-AT-GA (Andriushchenko & Flammarion, 2020).

**Remark 1** *Choice of learning rate for IG-involved descent term: In (11), the choice of* $\alpha_2$ *could affect the trade-off between accuracy and robustness (see empirical justification in Appendix E). Clearly, if* $\alpha_2 = 0$*, then the upper-level model parameter updating step reduces to the standard* FAST-AT*. In Sec. 5.1, we will show that the* $\alpha_2$*-associated term plays a positive role in alleviating catastrophic robust overfitting. Meanwhile,* $\lambda$ *in (7) could also affect the accuracy-robustness tradeoff. For example, if* $\lambda \to \infty$*, then* $\boldsymbol{\delta} = \mathbf{0}$ *(no robustness gain). Spurred by above, we choose the following combination of* $\alpha_2$ *and* $\lambda$*,* $\alpha_2/\lambda = 0.1\alpha_1$*, which works well in practice; see Table A1.*

**Remark 2** *Choice of linearization point* $\mathbf{z}$*: To specify (7), we investigate two classes of linearization schemes. The first class is random constant linearization, which includes: "uniformly random*

*linearization", i.e., $\mathbf{z} = \boldsymbol{\delta}_0$ as FAST-AT, and "random corner linearization" under the $\epsilon$-radius $\ell_\infty$-ball, i.e., $\mathbf{z} \in \{-\epsilon, \epsilon\}^d$. The second class is 1-step perturbation warm-up-based linearization, which includes the other two specifications: "1-step PGD" $\mathbf{z} = P_\mathcal{C}\left(\boldsymbol{\delta}_0 + \alpha \cdot \mathrm{sign}\left(\nabla_{\boldsymbol{\delta}}\ell_{\mathrm{tr}}(\boldsymbol{\theta}_t, \boldsymbol{\delta}_0)\right)\right)$, and "1-step PGD w/o sign" $\mathbf{z} = P_\mathcal{C}\left(\boldsymbol{\delta}_0 + \alpha \nabla_{\boldsymbol{\delta}}\ell_{\mathrm{tr}}(\boldsymbol{\theta}_t, \boldsymbol{\delta}_0)\right)$. We consider the aforementioned linearization schemes since FAST-BAT combined with these linearizations takes computation cost comparable to the baselines FAST-AT and FAST-AT-GA. Our experiments show that FAST-BAT using "1-step PGD w/o sign" leads to the best defense performance; see justification in Table A3.*

## 5 EXPERIMENTS

### 5.1 EXPERIMENT SETUP

**Datasets and model architectures.** We will evaluate the effectiveness of our proposal under CIFAR-10 (Krizhevsky & Hinton, 2009) and ImageNet (Deng et al., 2009). Unless specified otherwise, we will train DNN models PreActResNet (PARN)-18 (He et al., 2016b) for CIFAR-10, and ResNet (RN)-50 (He et al., 2016a) for ImageNet. As a part of ablation study, we also train larger models PARN-50 and WideResNet (WRN)-16-8 (Zagoruyko & Komodakis, 2016) on CIFAR-10.

**Baselines.** We consider three methods as our baselines: FAST-AT (Wong et al., 2020), FAST-AT-GA (Andriushchenko & Flammarion, 2020), and PGD-2-AT (Madry et al., 2018), *i.e.*, the 2-step PGD attack-based AT. The primal criterion of baseline selection is computation complexity. The training time of all methods including ours falls between the time of FAST-AT and that of FAST-AT-GA. We remark that when evaluating on ImageNet, we only compare ours with FAST-AT since as shown in Table 6 of (Andriushchenko & Flammarion, 2020), the other baseline methods did not show improvement over Fast-AT at the attack budget $\epsilon = 2/255$.

**Training details.** We choose the training perturbation strength $\epsilon \in \{2, 4, \ldots, 16\}/255$ for CIFAR-10 and $\epsilon = 2/255$ for ImageNet following (Wong et al., 2020; Andriushchenko & Flammarion, 2020). Throughout the experiments, we utilize an SGD optimizer with a momentum of $0.9$ and weight decay of $5 \times 10^{-4}$. For CIFAR-10, we train each model for 20 epochs in total, where we use cyclic scheduler to adjust the learning rate. The learning rate linearly ascends from 0 to 0.2 within the first 10 epochs and then reduces to 0 within the last 10 epochs. Our batch size is set to 128 for all settings. In the implementation of FAST-BAT, we adjust the hyperparameter $\lambda$ from $255/5000$ to $255/2000$ based on the specification of train-time $\epsilon$. For ImageNet, we strictly follows the setup given by Wong et al. (2020). In FAST-BAT, we set $\lambda = 255/3000$. For each method, we use the early stopping method to pick the model with best robust accuracy, following (Rice et al., 2020). All the CIFAR-10 experiments are conducted on a single Tesla P-100 GPU and all ImageNet experiments run on a single machine with two Tesla P-100s. All the baselines are implemented using the recommended training configurations in their official GitHub repos. We refer readers to Appendix D for more details on training setup.

**Evaluation details.** For adversarial evaluation, we report robust test accuracy (**RA**) of a learned model against PGD attacks (Madry et al., 2018) (**RA-PGD**). Unless otherwise specified, we set the test-time perturbation strength ($\epsilon$) same as the train-time value, and take 50-step PGD with 10 restarts for both CIFAR-10 and ImageNet evaluation. We also measure robust accuracy against AutoAttacks (Croce & Hein, 2020), termed **RA-AA**. Further, we measure the standard accuracy (**SA**) against natural examples. Results are averaged over 5 independent trials with different random seeds.

### 5.2 RESULTS

**Overall performance of FAST-BAT.** In Table 2 and 3, we compare the performance of our proposed FAST-BAT with baselines on CIFAR-10 and ImageNet, respectively.

Table 3: SA and RA on ImageNet.

| Method | SA (%) | RA-PGD (%) |
|---|---|---|
| FAST-AT | 60.90 | 43.43 |
| FAST-BAT | 60.18 | 44.64 |

*First*, we find that FAST-BAT consistently outperforms the other baselines across datasets and attack types. For example, FAST-BAT at least improves 1.35% RA-PGD and 1.41% RA-AA with test-time $\epsilon = 8/255$ in the training setup (CIFAR-10, $\epsilon = 8/255$). On ImageNet, FAST-BAT outperforms FAST-AT by 1.23% when facing attacks with $\epsilon = 2/255$.

Table 2: SA, RA-PGD and RA-AA of different robust training methods in the setup (CIFAR-10, PARN-18 training with $\epsilon = 8/255$) and (CIFAR-10, PARN-18 training with $\epsilon = 16/255$), respectively. All the results are averaged over 5 independent trials with different random seeds.

| | | RA-PGD (%) | | | | RA-AA (%) | | |
|---|---|---|---|---|---|---|---|---|
| Method | SA (%) | $\epsilon = 4$ | $\epsilon = 8$ | $\epsilon = 12$ | $\epsilon = 16$ | $\epsilon = 2$ | $\epsilon = 8$ | $\epsilon = 16$ |
| *CIFAR-10, PARN-18 trained with $\epsilon = 8/255$* | | | | | | | | |
| Fast-AT | 81.89±0.31 | 65.92 ±0.11 | 45.44 ±0.38 | 23.69 ±0.34 | 9.56 ±0.26 | 72.54 ±0.20 | 41.95 ±0.13 | 7.91 ±0.06 |
| Fast-AT-GA | 79.78±0.47 | 65.74 ±0.19 | 47.32 ±0.35 | 28.67 ±0.26 | 11.57 ±0.32 | 71.60 ±0.39 | 43.45 ±0.27 | 9.48 ±0.15 |
| PGD-2-AT | **83.26**±0.28 | 65.59 ±0.34 | 44.71 ±0.42 | 23.67 ±0.35 | 9.42 ±0.33 | **73.28**±0.15 | 41.73 ±0.20 | 7.54 ±0.25 |
| Fast-BAT | 79.47 ±0.14 | **66.26** ±0.08 | **48.67** ±0.18 | **29.87** ±0.46 | **14.00** ±0.21 | 72.07 ±0.22 | **44.86** ±0.34 | **11.51** ±0.20 |
| *CIFAR-10, PARN-18 trained with $\epsilon = 16/255$* | | | | | | | | |
| Fast-AT | 46.13±2.25 | 42.74 ±0.91 | 37.17 ±0.74 | 27.99 ±0.72 | 21.92 ±0.71 | 36.31 ±2.20 | 31.66 ±0.27 | 12.48 ±0.29 |
| Fast-AT-GA | 58.53 ±1.20 | 51.71 ±0.99 | 43.86 ±0.67 | 35.46 ±0.36 | **26.29** ±0.14 | 53.61 ±1.10 | 38.69 ±0.56 | 18.11 ±0.36 |
| PGD-2-AT | **69.40** ±0.30 | 59.25 ±0.16 | 48.79 ±0.31 | 32.12 ±5.63 | 24.30 ±0.46 | 61.90 ±0.28 | 41.59 ±0.22 | 15.40 ±0.29 |
| Fast-BAT | 67.81 ±0.18 | **59.35** ±0.13 | **49.05** ±0.12 | **37.71** ±0.36 | 26.07 ±0.28 | **62.16** ±0.14 | **43.64** ±0.26 | **18.18** ±0.34 |

*Second*, Fast-BAT leads to a better SA-RA trade-off compared with the other baselines. For example, in the setup of (CIFAR-10, PARN-18 trained with $\epsilon = 8/255$), we observe that Fast-BAT outperforms Fast-AT-GA in RA, without losing SA. And in the setup of (CIFAR-10, PARN-18 trained with $\epsilon = 16/255$), Fast-BAT significantly outperforms Fast-AT-GA, with 9.28% SA improvement and comparable or even better RA. Compared with PGD-2-AT, Fast-BAT is much more resilient against strong adversaries, *e.g.*, $\epsilon = 16$.

*Third*, the robustness advantage of our method becomes more notable when the test-time attack budget becomes smaller than the train-time budget. For examples, the RA-AA improvement of Fast-BAT over Fast-AT-GA grows from 0.07% (evaluated at $\epsilon = 16/255$) to 4.95% (evaluated at $\epsilon = 8/255$), and 8.55% (evaluated at $\epsilon = 2/255$) in the case of (CIFAR-10, trained with $\epsilon = 16/255$).

**Performance under different model architectures.** Besides PARN-18 reported above, Table 4 presents experiment results on both deeper (PARN-50) and wider (WRN-18-6) models. As we can see, Fast-BAT consistently yields RA improvement over the other baselines. We also note that PGD-2-AT could be a competitive baseline in terms of SA, *e.g.*, the case of (PARN-50, $\epsilon = 8/255$). In contrast to Fast-AT and Fast-AT-GA, Fast-BAT is the only approach that yields an evident RA improvement over PGD-2-AT.

Table 4: Performance of different robust training methods under different model types. All the models are both trained and evaluated with the same perturbation strength $\epsilon$.

| Model | Method | SA(%) ($\epsilon = 8/255$) | RA-PGD(%) ($\epsilon = 8/255$) | SA(%) ($\epsilon = 16/255$) | RA-PGD(%) ($\epsilon = 16/255$) |
|---|---|---|---|---|---|
| PARN-50 | Fast-AT | 73.15±6.10 | 41.03±2.99 | 43.86±4.31 | 22.08±0.27 |
| | Fast-AT-GA | 77.40±0.81 | 46.16±0.98 | 42.28±6.69 | 22.87±1.25 |
| | PGD-2-AT | **83.53**±0.17 | 46.17±0.59 | 68.88±0.39 | 22.37±0.41 |
| | Fast-BAT | 78.91±0.68 | **49.18**±0.35 | **69.01**±0.19 | **24.55**±0.06 |
| WRN-16-8 | Fast-AT | 84.39±0.46 | 45.80±0.57 | 49.39±2.17 | 21.99±0.41 |
| | Fast-AT-GA | 81.51±0.38 | 48.29±0.20 | 45.95±13.65 | 23.10±3.90 |
| | PGD-2-AT | **85.52**±0.14 | 45.47±0.14 | **72.11**±0.33 | 23.61±0.16 |
| | Fast-BAT | 81.66±0.54 | **49.93**±0.36 | 68.12±0.47 | **25.63**±0.44 |

**Mitigation of robustness catastrophic overfitting.** As shown in (Andriushchenko & Flammarion, 2020), Fast-AT suffers robustness catastrophic overfitting when the train-time and test-time attack strength $\epsilon$ grows. Following (Andriushchenko & Flammarion, 2020), Figure 1 presents two RA-PGD trajectories, *i.e.*, training w/o early stopping and training w/ early stopping, versus the train- and test-time $\epsilon$. As

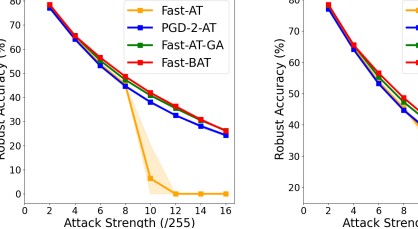
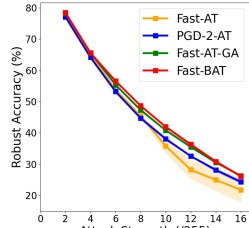

(a) Without early stopping    (b) With early stopping

Figure 1: RA-PGD of different robust training methods for (CIFAR-10, PARN-18) with the same training and evaluation attack strengths.

we can see, Fast-AT encounters a sharp RA drop when $\epsilon > 8$ when early stopping is not used, consistent with (Andriushchenko & Flammarion, 2020). Assisted by early stopping, the overfitting

of RA can be alleviated to some extent for FAST-AT, but its performance still remains the worst. Moreover, different from (Andriushchenko & Flammarion, 2020), we find that PGD-2-AT yields resilient performance against robustness catastrophic overfitting. Our implementation gives a more positive baseline than the implementation of PGD-2-AT in (Andriushchenko & Flammarion, 2020), since the latter did not use random initialization to generate train-time attacks. Furthermore, Figure 1 shows that our proposal mitigates the issue of robustness catastrophic overfitting and yields improved RA over the other baselines. We highlight that such a achievement made by FAST-BAT is 'free' of any robustness stability regularization, like gradient alignment used in FAST-AT-GA.

**Gradient alignment for 'free'.** As shown by Andriushchenko & Flammarion (2020), gradient alignment (GA) is a key performance indicator to measure the appearance of robustness catastrophic overfitting. The insight from Figure 1 suggested that FAST-BAT can mitigate overfitting without using explicit GA regularization. Spurred by above, Figure 2 presents the GA score versus the training epoch number, where GA characterizes the sensitivity of loss landscape against random input perturbations; see derivations in (Andriushchenko & Flammarion, 2020). The higher GA is, the more stable the robust training is. Figure 2 shows that FAST-BAT automatically enforces

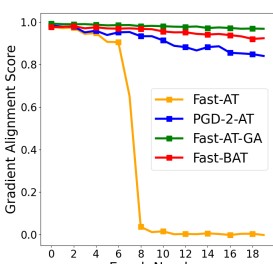

Figure 2: GA evaluation.

GA and it outperforms FAST-AT and PGD-2-AT. FAST-BAT remains very close to FAST-AT-GA, which maximizes GA using an extra train-time regularization The above empirical results imply that gradient alignment may be just a necessary condition for avoiding catastrophic overfitting, but not a sufficient one. A possible justification can be made from the perspective of the flatness of the loss landscape. A higher gradient alignment implies a flatter loss landscape with respect to input perturbations. However, the direct penalization on the norm of the input gradient may not achieve the state-of-the-art model robustness.

**Sanity check for obfuscated gradients** As pointed out by Athalye et al. (2018a), model robustness could be overestimated due to obfuscated gradients. The model with obfuscated gradients could have 'obfuscated' stronger resilience to white-box attacks than transfer (black-box) attacks. To justify the validity of FAST-BAT, Table 2 summarizes the comparison between our

Table 5: RA of robust PARN-18 trained by the four different methods against adaptive attacks ('RA-PGD' column) and transfer attacks ('Transfer Attack' columns). Naturally trained PARN-18, PARN-50, and WRN-16-8 are used as surrogate models for PGD-20 attack with $\epsilon = 8/255$.

| Method | RA-PGD(%) | RA-Transfer Attack(%) | | |
|---|---|---|---|---|
| | | PARN-18 | PARN-50 | WRN-16-8 |
| FAST-AT | 45.44 | 76.35 | 76.94 | 77.23 |
| PGD-2-AT | 44.71 | 77.56 | 78.64 | 78.84 |
| FAST-AT-GA | 47.31 | 77.34 | 78.34 | 78.53 |
| **FAST-BAT(Ours)** | **48.67** | **78.03** | **79.93** | **79.21** |

proposal and the other baselines when facing white-box adaptive and black-box transfer attacks. Firstly, RA increases if the transfer attack is present for each method, implying that the transfer attack is weaker than the white-box adaptive attack. This is desired in the absence of obfuscated gradients. Moreover, FAST-BAT consistently outperforms the other three baselines when defending against both adaptive and transfer attacks. The absence of obfuscated gradients can also be justified by RA vs. the growth of attack budget $\epsilon$ in Table 2, and the flatness of adversarial loss landscape in Figure. A1.

**Ablation studies.** In Appendix E, we present additional empirical studies including 1) the sensitivity analysis of the linearization hyperparameter $\lambda$, 2) the choice of the linearization point, 3) the sensitivity analysis of $\alpha_2$, 4) the influence of Hessian matrix on ReLU, and 5) non-ReLU neural networks.

## 6 CONCLUSION

In this paper, we introduce a novel bi-level optimization (BLO)-based fast adversarial training framework, termed FAST-BAT. The rationale behind designing fast robust training through the lens of BLO lies in two aspects. First, from the perspective of implicit gradients, we show that existing FAST-AT framework is equivalent to the lower-level linearized BLO along the sign direction of input gradient. Second, we show that FAST-BAT enables the least restriction to achieve improved staibility of performance, mitigated catastrophic overfitting, and enhanced accuracy-robustness trade-off. To the best of our knowledge, we for the first time establish the theory and the algorithmic foundation of BLO for adversarially robust training. Extensive experiments are provided to demonstrate the superiority of our method to state-of-the-art accelerated AT baselines.

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

# A    PROOF OF THEOREM 1

**Proof**: Upon defining $g(\boldsymbol{\theta}, \boldsymbol{\delta}) = (\boldsymbol{\delta} - \mathbf{z})^\top \nabla_{\boldsymbol{\delta} = \mathbf{z}} \ell_{\mathrm{atk}}(\boldsymbol{\theta}, \boldsymbol{\delta}) + (\lambda/2)\|\boldsymbol{\delta} - \mathbf{z}\|_2^2$, we repeat (7) as

$$
\begin{aligned}
\underset{\boldsymbol{\theta}}{\text{minimize}} \quad & \mathbb{E}_{(\mathbf{x}, y) \in \mathcal{D}}[\ell_{\mathrm{tr}}(\boldsymbol{\theta}, \boldsymbol{\delta}^*(\boldsymbol{\theta}))] \\
\text{subject to} \quad & \boldsymbol{\delta}^*(\boldsymbol{\theta}) = \underset{\mathbf{B}\boldsymbol{\delta} \leq \mathbf{b}}{\arg\min}\, g(\boldsymbol{\theta}, \boldsymbol{\delta}),
\end{aligned}
\tag{12}
$$

where we have used the expression of linear constraints in (9).

Our goal is to derive the IG $\frac{d\boldsymbol{\delta}^*(\boldsymbol{\theta})^\top}{d\boldsymbol{\theta}}$ shown in (3). To this end, we first build implicit functions by leveraging KKT conditions of the lower-level problem of (12). We say $\boldsymbol{\delta}^*(\boldsymbol{\theta})$ and $\boldsymbol{\lambda}^*(\boldsymbol{\theta})$ (Lagrangian multipliers) satisfy the KKT conditions:

$$
\begin{aligned}
\text{Stationarity:} \quad & \nabla_{\boldsymbol{\delta}} g(\boldsymbol{\theta}, \boldsymbol{\delta}^*(\boldsymbol{\theta})) + \mathbf{B}^\top \boldsymbol{\lambda}^*(\boldsymbol{\theta}) = \mathbf{0}, \\
\text{Complementary slackness :} \quad & \boldsymbol{\lambda}^*(\boldsymbol{\theta}) \cdot (\mathbf{B}\boldsymbol{\delta}^*(\boldsymbol{\theta}) - \mathbf{b}) = \mathbf{0} \\
\text{Dual feasibility:} \quad & \boldsymbol{\lambda}^*(\boldsymbol{\theta}) \geq \mathbf{0}
\end{aligned}
\tag{13}
$$

where $\cdot$ denotes the elementwise product.

*Active constraints and definition of* $\mathbf{B}_0$: Let $\mathbf{B}_0$ denote the sub-matrix of $\mathbf{B}$ and $\mathbf{b}_0$ the sub-vector of $\mathbf{b}$, which consists of only the *active constraints* at $\boldsymbol{\delta}^*(\boldsymbol{\theta})$, *i.e.*, those satisfied with the equality $\mathbf{B}_0\boldsymbol{\delta}^*(\boldsymbol{\theta}) = \mathbf{b}_0$ (corresponding to *nonzero* dual variables). The determination of active constraints is done given $\boldsymbol{\theta}$ at each iteration.

With the aid of $(\mathbf{B}_0, \mathbf{b}_0)$, KKT (13) becomes

$$
\nabla_{\boldsymbol{\delta}} g(\boldsymbol{\theta}, \boldsymbol{\delta}^*(\boldsymbol{\theta})) + \mathbf{B}_0^\top \boldsymbol{\lambda}^*(\boldsymbol{\theta}) = \mathbf{0}, \quad \text{and} \quad \mathbf{B}_0 \boldsymbol{\delta}^*(\boldsymbol{\theta}) - \mathbf{b}_0 = \mathbf{0},
\tag{14}
$$

where the nonzero $\boldsymbol{\lambda}^*(\boldsymbol{\theta})$ only correspond to active constraints. We take derivatives w.r.t. $\boldsymbol{\theta}$ for (14), and thus obtain

$$
\frac{d\nabla_{\boldsymbol{\delta}} g(\boldsymbol{\theta}, \boldsymbol{\delta}^*(\boldsymbol{\theta}))^\top}{d\boldsymbol{\theta}} + \nabla_{\boldsymbol{\theta}} \boldsymbol{\lambda}^*(\boldsymbol{\theta})^\top \mathbf{B}_0 = \mathbf{0}
$$

$$
\Longrightarrow \nabla_{\boldsymbol{\theta}\boldsymbol{\delta}} g(\boldsymbol{\theta}, \boldsymbol{\delta}^*(\boldsymbol{\theta})) + \underbrace{\frac{d\boldsymbol{\delta}^*(\boldsymbol{\theta})^\top}{d\boldsymbol{\theta}}}_{\text{IG}} \nabla_{\boldsymbol{\delta}\boldsymbol{\delta}} g(\boldsymbol{\theta}, \boldsymbol{\delta}^*(\boldsymbol{\theta})) + \nabla_{\boldsymbol{\theta}} \boldsymbol{\lambda}^*(\boldsymbol{\theta})^\top \mathbf{B}_0 = \mathbf{0},
\tag{15}
$$

$$
\text{and} \quad \underbrace{\frac{d\boldsymbol{\delta}^*(\boldsymbol{\theta})^\top}{d\boldsymbol{\theta}}}_{\text{IG}} \mathbf{B}_0^\top = \mathbf{0},
\tag{16}
$$

where $\nabla_{\boldsymbol{\theta}\boldsymbol{\delta}} \in \mathbb{R}^{|\boldsymbol{\theta}| \times |\boldsymbol{\delta}|}$ denotes second-order partial derivatives (recall that $|\boldsymbol{\theta}| = n$ and $|\boldsymbol{\delta}| = d$). According to (15), we have

$$
\frac{d\boldsymbol{\delta}^*(\boldsymbol{\theta})^\top}{d\boldsymbol{\theta}} = -[\nabla_{\boldsymbol{\theta}\boldsymbol{\delta}} g(\boldsymbol{\theta}, \boldsymbol{\delta}^*(\boldsymbol{\theta})) + \nabla_{\boldsymbol{\theta}} \boldsymbol{\lambda}^*(\boldsymbol{\theta})^\top \mathbf{B}_0] \nabla_{\boldsymbol{\delta}\boldsymbol{\delta}} g(\boldsymbol{\theta}, \boldsymbol{\delta}^*(\boldsymbol{\theta}))^{-1}.
\tag{17}
$$

Substituting the above into (16), we obtain

$$
\nabla_{\boldsymbol{\theta}\boldsymbol{\delta}} g(\boldsymbol{\theta}, \boldsymbol{\delta}^*(\boldsymbol{\theta})) \nabla_{\boldsymbol{\delta}\boldsymbol{\delta}} g(\boldsymbol{\theta}, \boldsymbol{\delta}^*(\boldsymbol{\theta}))^{-1} \mathbf{B}_0^\top + \nabla_{\boldsymbol{\theta}} \boldsymbol{\lambda}^*(\boldsymbol{\theta})^\top \mathbf{B}_0 \nabla_{\boldsymbol{\delta}\boldsymbol{\delta}} g(\boldsymbol{\theta}, \boldsymbol{\delta}^*(\boldsymbol{\theta}))^{-1} \mathbf{B}_0^\top = \mathbf{0},
\tag{18}
$$

which yields:

$$
\nabla_{\boldsymbol{\theta}} \boldsymbol{\lambda}^*(\boldsymbol{\theta})^\top = -\nabla_{\boldsymbol{\theta}\boldsymbol{\delta}} g(\boldsymbol{\theta}, \boldsymbol{\delta}^*(\boldsymbol{\theta})) \nabla_{\boldsymbol{\delta}\boldsymbol{\delta}} g(\boldsymbol{\theta}, \boldsymbol{\delta}^*(\boldsymbol{\theta}))^{-1} \mathbf{B}_0^\top [\mathbf{B}_0 \nabla_{\boldsymbol{\delta}\boldsymbol{\delta}} g(\boldsymbol{\theta}, \boldsymbol{\delta}^*(\boldsymbol{\theta}))^{-1} \mathbf{B}_0^\top]^{-1},
\tag{19}
$$

and thus,

$$
\nabla_{\boldsymbol{\theta}} \boldsymbol{\lambda}^*(\boldsymbol{\theta})^\top \mathbf{B}_0 = -\nabla_{\boldsymbol{\theta}\boldsymbol{\delta}} g(\boldsymbol{\theta}, \boldsymbol{\delta}^*(\boldsymbol{\theta})) \nabla_{\boldsymbol{\delta}\boldsymbol{\delta}} g(\boldsymbol{\theta}, \boldsymbol{\delta}^*(\boldsymbol{\theta}))^{-1} \mathbf{B}_0^\top [\mathbf{B}_0 \nabla_{\boldsymbol{\delta}\boldsymbol{\delta}} g(\boldsymbol{\theta}, \boldsymbol{\delta}^*(\boldsymbol{\theta}))^{-1} \mathbf{B}_0^\top]^{-1} \mathbf{B}_0.
\tag{20}
$$

Substituting (20) into (17), we obtain the IG

$$
\begin{aligned}
\frac{d\boldsymbol{\delta}^*(\boldsymbol{\theta})^\top}{d\boldsymbol{\theta}} &= -\nabla_{\boldsymbol{\theta}\boldsymbol{\delta}} g(\boldsymbol{\theta}, \boldsymbol{\delta}^*(\boldsymbol{\theta})) \nabla_{\boldsymbol{\delta}\boldsymbol{\delta}} g(\boldsymbol{\theta}, \boldsymbol{\delta}^*(\boldsymbol{\theta}))^{-1} - \nabla_{\boldsymbol{\theta}} \boldsymbol{\lambda}^*(\boldsymbol{\theta})^\top \mathbf{B}_0 \nabla_{\boldsymbol{\delta}\boldsymbol{\delta}} g(\boldsymbol{\theta}, \boldsymbol{\delta}^*(\boldsymbol{\theta}))^{-1} \\
&= -\nabla_{\boldsymbol{\theta}\boldsymbol{\delta}} g(\boldsymbol{\theta}, \boldsymbol{\delta}^*(\boldsymbol{\theta})) \nabla_{\boldsymbol{\delta}\boldsymbol{\delta}} g(\boldsymbol{\theta}, \boldsymbol{\delta}^*(\boldsymbol{\theta}))^{-1} \\
&\quad + \nabla_{\boldsymbol{\theta}\boldsymbol{\delta}} g(\boldsymbol{\theta}, \boldsymbol{\delta}^*(\boldsymbol{\theta})) \nabla_{\boldsymbol{\delta}\boldsymbol{\delta}} g(\boldsymbol{\theta}, \boldsymbol{\delta}^*(\boldsymbol{\theta}))^{-1} \mathbf{B}_0^\top [\mathbf{B}_0 \nabla_{\boldsymbol{\delta}\boldsymbol{\delta}} g(\boldsymbol{\theta}, \boldsymbol{\delta}^*(\boldsymbol{\theta}))^{-1} \mathbf{B}_0^\top]^{-1} \mathbf{B}_0 \nabla_{\boldsymbol{\delta}\boldsymbol{\delta}} g(\boldsymbol{\theta}, \boldsymbol{\delta}^*(\boldsymbol{\theta}))^{-1}.
\end{aligned}
\tag{21}
$$

To further compute (21), the Hessian matrix $\nabla_{\boldsymbol{\delta\delta}}\ell_{\mathrm{atk}}$ is needed. Recall from the definition of the lower-level objective that the Hessian matrix is given by

$$\nabla_{\boldsymbol{\delta\delta}}g(\boldsymbol{\theta},\boldsymbol{\delta}^*(\boldsymbol{\theta})) = \nabla_{\boldsymbol{\delta\delta}}\ell_{\mathrm{atk}} + \lambda\mathbf{I} = \mathbf{0} + \lambda\mathbf{I}. \tag{22}$$

Here we used the assumption that $\nabla_{\boldsymbol{\delta\delta}}\ell_{\mathrm{atk}} = \mathbf{0}$. The rationale behind that is neural networks commonly leads to a piece-wise linear decision boundary w.r.t. the inputs (Moosavi-Dezfooli et al., 2019; Alfarra et al., 2020), and thus, its second-order derivative (Hessian) $\nabla_{\boldsymbol{\delta\delta}}\ell_{\mathrm{atk}}$ is close to zero.

Based on the simplification (22), we have

$$\frac{d\boldsymbol{\delta}^*(\boldsymbol{\theta})^\top}{d\boldsymbol{\theta}} = -(1/\lambda)\nabla_{\boldsymbol{\theta\delta}}g(\boldsymbol{\theta},\boldsymbol{\delta}^*(\boldsymbol{\theta}))\underbrace{\left(\mathbf{I} - \mathbf{B}_0^\top[\mathbf{B}_0\mathbf{B}_0^\top]^{-1}\mathbf{B}_0\right)}_{:=\mathbf{H}_{\mathcal{C}}}$$

$$-(1/\lambda)\nabla_{\boldsymbol{\theta\delta}}\ell_{\mathrm{atk}}(\boldsymbol{\theta},\boldsymbol{\delta}^*(\boldsymbol{\theta}))\mathbf{H}_{\mathcal{C}}, \tag{23}$$

where we have used the fact that $\nabla_{\boldsymbol{\theta\delta}}g = \nabla_{\boldsymbol{\theta\delta}}\ell_{\mathrm{atk}}$.

What is $\mathbf{H}_{\mathcal{C}}$ in (23)? Since $\mathbf{B} = \begin{bmatrix}\mathbf{I} \\ -\mathbf{I}\end{bmatrix}$, we can obtain that $\mathbf{B}_0\mathbf{B}_0^\top = \mathbf{I}$ and $\mathbf{B}_0^\top\mathbf{B}_0$ is a sparse diagonal matrix with diagonal entries being 0 or 1. Thus, $\mathbf{H}_{\mathcal{C}}$ can be first simplified to

$$\mathbf{H}_{\mathcal{C}} = \mathbf{I} - \mathbf{B}_0^\top\mathbf{B}_0. \tag{24}$$

Clearly, $\mathbf{H}_{\mathcal{C}}$ is also a diagonal matrix with either 0 or 1 diagonal entries. The 1-valued diagonal entry of $\mathbf{H}_{\mathcal{C}}$ corresponds to the *inactive constraints* in $\mathbf{B}\boldsymbol{\delta}^*(\boldsymbol{\theta}) < \mathbf{b}$, *i.e.*, those satisfied with *strict inequalities* in $\{\|\boldsymbol{\delta}\|_\infty \le \epsilon, \mathbf{0} \le \boldsymbol{\delta} \le \mathbf{1}\}$. This can be expressed as

$$\mathbf{H}_{\mathcal{C}} = \left[1_{p_1 \le \delta_1^* \le q_1}\mathbf{e}_1, \ldots, 1_{p_1 \le \delta_d^* \le q_d}\mathbf{e}_d\right] \tag{25}$$

where $1_{p_i \le \delta_i^* \le q_i} \in \{0,1\}$ denotes the indicator function over the constraint $\{p_i \le \delta_i^* \le q_i\}$ and returns 1 if the constraint is satisfied, $\delta_i^*$ denotes the $i$th entry of $\boldsymbol{\delta}^*(\boldsymbol{\theta})$, $p_i = \max\{-\epsilon, -x_i\}$ and $q_i = \min\{\epsilon, 1 - x_i\}$, and $\mathbf{e}_i \in \mathbb{R}^d$ denotes the basis vector with the $i$th entry being 1 and others being 0s.

Based on the definition of $g$, (23) and (25), we can eventually achieve the desired IG formula (10). The proof is now complete. $\qquad\square$

## B   DISCUSSION ON CASE $\ell_{\text{atk}} = -\ell_{\text{tr}}$

We provide an explanation on the argument "Even if we set $\ell_{\text{atk}} = -\ell_{\text{tr}}$, problem (2) does not reduce to problem (1) due to the presence of lower-level constraint" from the following two points.

- In the **absence** of the constraint $\boldsymbol{\delta} \in \mathcal{C}$, if we set $\ell_{\text{atk}} = -\ell_{\text{tr}}$, then Problem 2 will reduce to Problem 1.

  This is a known BLO result (*e.g.* Ghadimi & Wang (2018)) and can be readily proven using the stationary condition. To be specific, based on the stationary condition of unconstrained lower-level optimization, we have $\nabla_{\boldsymbol{\delta}} \ell_{\text{atk}}(\boldsymbol{\theta}, \boldsymbol{\delta}^*) = 0$. Since $\ell_{\text{atk}} = -\ell_{\text{tr}}$, we have $\nabla_{\boldsymbol{\delta}} \ell_{\text{tr}}(\boldsymbol{\theta}, \boldsymbol{\delta}^*) = 0$. As a result, the second term in Eq. 3 becomes $\mathbf{0}$ and solving problem 2 becomes identical to solving the min-max problem 1.

- In the **presence** of the constraint $\boldsymbol{\delta} \in \mathcal{C}$, the stationary condition cannot be applied since the stationary point may not be a feasible point in the constraint. In other words, $\nabla_{\boldsymbol{\delta}} \ell_{\text{atk}}(\boldsymbol{\theta}, \boldsymbol{\delta}^*) = 0$ does not hold in the case of $\ell_{\text{atk}} = -\ell_{\text{tr}}$. As a matter of fact, one has to resort to KKT conditions instead of the stationary condition for a constrained lower-level problem. Similar to our proof in Theorem 1, the implicit gradient (and thus the second term of Eq. 3) cannot be omitted in general. This makes problem 2 different from the problem 1.

## C   DERIVATION OF IMPLICIT GRADIENT FOR FAST-AT

We can derive Eq. 6 using KKT condition similar to Theorem 1. Specifically, let

$$g(\boldsymbol{\theta}, \boldsymbol{\delta}) = \langle \text{sign}(\nabla_{\boldsymbol{\delta}=\mathbf{z}} \ell_{\text{atk}}(\boldsymbol{\theta}, \boldsymbol{\delta}; \mathbf{x}, y)), \boldsymbol{\delta} - \mathbf{z} \rangle + \frac{\lambda}{2} \|\boldsymbol{\delta} - \mathbf{z}\|_2^2, \tag{26}$$

we have

$$\nabla_{\boldsymbol{\theta}\boldsymbol{\delta}} g = \mathbf{0}. \tag{27}$$

Following (21), we can further obtain (6) based on (27).

## D   DETAILED EXPERIMENT SETTINGS

### D.1   TRAINING SET-UP

For CIFAR-10, we summarize the training setup for each method. 1) FAST-AT: We use FGSM with an attack step size of $1.25\epsilon$ to generate perturbations; 2) PGD-2-AT: 2-step PGD attacks[1] with an attack step size of $0.5\epsilon$ is implemented; 3) FAST-AT-GA: The gradient alignment regularization parameter is set to the recommended value for each $\epsilon$; 4) FAST-BAT: We select $\lambda$ from $255/5000$ to $255/2000$ for different $\epsilon$. At the same time, we adjust $\alpha_2$ accordingly, so that the coefficient of the second term in (11), namely $\alpha_2/\lambda$ always equals to $0.1\alpha_1$.

For ImageNet, we set $\epsilon$ to $2/255$, and we strictly follow the training setting adopted by Wong et al. (2020). In FAST-BAT, we fix $\lambda$ at $255/3000$ and adopt the same $\alpha_2$ selection strategy as CIFAR-10.

**Parameter for FAST-AT-GA**   Regarding FAST-AT-GA with different model types, we adopt the same regularization parameter recommended in its official repo[2] intended for PreActResNet-18 (namely 0.2 for $\epsilon = 8/255$ and 2.0 for $\epsilon = 16/255$).

---

[1]We use random initialization to generate perturbations for PGD, while in the paper of FAST-AT-GA (Andriushchenko & Flammarion, 2020), 2-step PGD is initialized at zero point, which we believe will underestimate the effect of PGD-2-AT

[2]FAST-AT-GA: https://github.com/tml-epfl/understanding-fast-adv-training/blob/master/sh

# E ADDITIONAL EXPERIMENTAL RESULTS

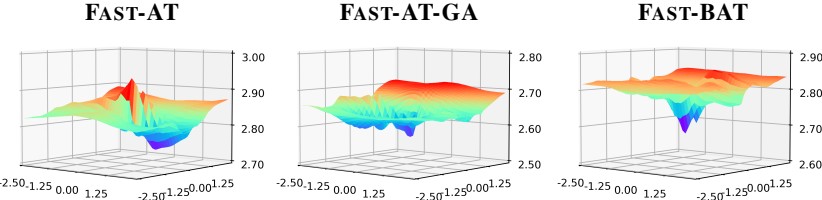

**FAST-AT**          **FAST-AT-GA**          **FAST-BAT**

Figure A1: Visualization of adversarial loss landscapes of FAST-AT, FAST-AT-GA and FAST-BAT trained using the ResNet-18 model on the CIFAR-10 dataset. The losses at are calculated w.r.t. the same image example ID #001456, and the landscape is obtained by tracking the loss changes w.r.t. input variations following Engstrom et al. (2018). That is, the loss landscape is generated by $z = \text{loss}(I + x \cdot \mathbf{r_1} + y \cdot \mathbf{r_2})$, where $I$ denotes an image, and the $x$-axis and the $y$-axis correspond to linear coefficients associated with the sign-based attack direction $\mathbf{r_1} = \text{sign}(\nabla_I \text{loss}(I)) \, \hat{\mathbf{x}}$ and a random direction $\mathbf{r_2} \sim \text{Rademacher}(0.5)$, respectively.

**Sensitivity to regularization parameter** $\lambda$
In Table A1, we show the sensitivity of FAST-BAT to the regularization parameter $\lambda$. All the parameters remain the same as default setting, except that for different $\lambda$. We always adjust $\alpha_2$ so that $\alpha_2/\lambda = 0.1\alpha_1$ holds. Note $1/\lambda$ also serves as the attack step in (8). As $\lambda$ decreases, the improvement on robust accuracy is evidently strengthened, and there is a obvious trade-off between robust accuracy (SA) and standard accuracy (RA). At a certain level of $\lambda$, namely when $\lambda \leq 255/2000$, RA starts to converge and stop surging.

Table A1: Performance of FAST-BAT with different parameter $\lambda$. We train and evaluate with the same attack budget $\epsilon = 16/255$ on CIFAR-10 to show the influence brought by $\lambda$.

| CIFAR-10, PreActResNet-18, $\epsilon = 16/255$ | | | | | |
|---|---|---|---|---|---|
| $1/\lambda$ (/255) | 500 | 1000 | 1500 | 2000 | 2500 |
| SA (%) | **83.20** | 75.06 | 69.31 | 67.81 | 67.59 |
| RA-PGD (%) | 19.02 | 21.42 | 23.34 | 26.07 | **26.12** |

**Sensitivity to different** $\alpha_2$ **choices**  We consider the case of robust training with the large $\epsilon$ choice (16/255). As we can see from Table A2, if $\alpha_2$ is set too small ($\alpha_2 = 0.008\alpha_1$), then both SA and RA will drop significantly. Here $\alpha_1$ is set as the cyclic learning rate and thus not a constant parameter. However, in the $\alpha_2$ interval $[0.0125\alpha_1, 0.025\alpha_1]$, we observed a tradeoff between standard accuracy (SA) and robust accuracy (RA): That is, the improvement in RA corresponds to a loss in SA. In our experiments, we choose $\alpha_2$ when the tradeoff yields the best RA without suffering a significant drop of SA (which still outperforms the baseline approaches).

Table A2: Performance of FAST-BAT with different $\alpha_2$ choices on CIFAR-10. Models are trained and evaluated with the same attack budget ($\epsilon = 16/255$). Here $\alpha_1$ is set as the cyclic learning rate and thus, is not a constant parameter. $\alpha_2$ is always set proportionate to $\alpha_1$ for simplicity.

| $\alpha_2$ (**CIFAR-10, PreActResNet18,** $\epsilon = 16/255$) | $0.025\alpha_1$ | $0.0167\alpha_1$ | $0.0125\alpha_1$ | $0.008\alpha_1$ |
|---|---|---|---|---|
| SA (%) | **75.06** | 69.31 | 67.81 | 57.92 |
| RA-PGD (%) | 21.42 | 23.34 | **26.07** | 20.53 |

**Sensitivity of linearization schemes**  Fast-BAT needs a good linearization point $\mathbf{z}$ in (7). In experiments, we adopt the perturbation generated by 1-step PGD without sign as our default linearization scheme. In Table A3, we show the performance of the other possible linearization options. We can find 1-step PGD without sign achieves best robust accuracy among all the choices. This is not spurring since this linearization point choice is consistent with the first-Taylor expansion that we used along the direction of input gradient without sign involved. By contrast, FAST-BAT linearized with uniformly random noise suffers from catastrophic overfitting and reaches a rather low standard accuracy (SA). FAST-BAT with other linearizations also yields worse SA-RA trade-off than our proposal.

Table A3: Performance of FAST-BAT with different linearization schemes. Besides 1-step PGD without sign (**PGD w/o Sign**), we further generate linearization point with the following methods: uniformly random noise $[-\epsilon, \epsilon]^d$ (**Uniformly Random**); uniformly random corner $\{-\epsilon, \epsilon\}^d$ (**Random Corner**); and perturbation from 1-step PGD attack with $0.5\epsilon$ as attack step (**PGD**).

| CIFAR-10, PreActResNet-18, $\epsilon = 16/255$ | | | | |
|---|---|---|---|---|
| Linearization Method | PGD w/o Sign | Uniformly Random | Random Corner | PGD |
| SA (%) | 69.31 | 43.42 | 62.19 | **75.30** |
| RA-PGD (%) | **23.34** | 21.25 | 16.5 | 19.42 |

Table A4: Performance of Hessian-free and Hessian-aware FAST-BATon CIFAR-10. We train and evaluate with the same attack budgets $\epsilon = 8/255$ and $\epsilon = 16/255$ to show the influence brought by Hessian matrix.

| Method | RA-PGD (%) ($\epsilon = 8/255$) | RA-PGD (%) ($\epsilon = 16/255$) | RA-AA (%) ($\epsilon = 8/255$) | RA-AA (%) ($\epsilon = 16/255$) | SA (%) ($\epsilon = 8/255$) | SA (%) ($\epsilon = 16/255$) | Time (s/epoch) |
|---|---|---|---|---|---|---|---|
| Hessian-free Fast-BAT | 48.67 | 26.07 | 44.86 | 18.18 | 79.47 | 67.81 | 135 |
| Hessian-aware Fast-BAT | 48.52 | 26.12 | 44.81 | 18.31 | 79.54 | 67.93 | 179 |

Table A5: Performance of FAST-ATand FAST-BATwith different activation functions on CIFAR-10. ReLU, Swish and Softplus are taken into consideration. For FAST-BAT, we compare the Hessian-free and Hessian-aware version to verify the influence of Hessian matrix. The results are averaged over 3 independent trials.

| Setting | SA (%) ($\epsilon = 8/255$) | RA-PGD (%) ($\epsilon = 8/255$) | SA (%) ($\epsilon = 16/255$) | RA-PGD (%) ($\epsilon = 16/255$) | Time (s/epoch) |
|---|---|---|---|---|---|
| Fast-AT-ReLU | 81.88 | 45.44 | 46.13 | 21.75 | 42 |
| Fast-BAT-ReLU (Hessian-aware) | 79.54 | 48.52 | 67.93 | 26.12 | 179 |
| Fast-BAT-ReLU (Hessian-free) | 79.47 | 48.67 | 67.81 | 26.07 | 135 |
| Fast-AT-Softplus | 81.29 | 47.26 | 45.39 | 22.40 | 42 |
| Fast-BAT-Softplus (Hessian-aware) | 79.59 | 49.74 | 68.63 | 25.54 | 178 |
| Fast-BAT-Softplus (Hessian-free) | 79.48 | 49.67 | 68.57 | 25.59 | 137 |
| Fast-AT-Swish | 75.61 | 44.43 | 52.03 | 23.08 | 49 |
| Fast-BAT-Swish (Hessian-aware) | 73.93 | 45.97 | 62.49 | 23.99 | 196 |
| Fast-BAT-Swish (Hessian-free) | 73.89 | 45.90 | 62.59 | 23.81 | 141 |

**Influence of Hessian matrix**    In Theorem 1, the Hessian-free assumption, *i.e.*$\nabla_{\delta\delta}\ell_{\mathrm{atk}} = 0$, was made to simplify the computation of IG term (implicit gradient). To examine how the Hessian matrix $\nabla_{\delta\delta}\ell_{\mathrm{atk}}$ affects the performance of Fast-BAT, we conduct experiments to compare the Hessian-free FAST-BATwith the Hessian-aware version. In Hessian-aware FAST-BAT, the implicit gradient is calculated based on (21). In Table A4, the results do not indicate much difference when Hessian is used. However, the extra calculation brought by Hessian heavily slows down FAST-BATas around 30% more time is needed. Therefore, the Hessian-free assumption is reasonable and also necessary in terms of the efficiency of the algorithm.

**Ablation study on smooth activation functions**    The Hessian-free assumption is based on the fact that the commonly used ReLU activation function is piece-wise linear *w.r.t.* input. We further conduct experiments to verify the feasibility of such assumption on models with non-ReLU activation functions. We choose two commonly used activation functions, Swish(Ramachandran et al., 2017) and Softplus, as alternatives for non-smooth ReLU function. We compare the results both calculating Hessian as well as the Hessian-free version to see if the Hessian-free assumption still holds for the non-ReLU neural network. The results are shown in Table A5. As we can see, the use of Hessian does not affect performance much. A similar phenomenon can be observed across different $\epsilon$ and different model activation functions (ReLU, Softplus, and Swish). However, the introduction of Hessian leads to an increase in time consumption by more than 30%. Therefore, we can draw the conclusion that the Hessian-free assumption is reasonable across different activation function choices.

