# OpenReview forum: "Revisiting and Advancing Fast Adversarial Training Through the lens of Bi-Level Optimization"
_ICLR.cc/2022/Conference — ICLR 2022 Submitted_

### Official Review · Reviewer_N6su · 2021-10-25

**Correctness:** 2
**Technical Novelty And Significance:** 3
**Empirical Novelty And Significance:** 3
**Recommendation:** 5
**Confidence:** 5

**Main Review:**

In general, I like this perspective due to the realizing of the implicit gradient term since actually the adversarial perturbation depends on the network weights, and this is also related to some versions of GANs’ training. However, in the current version of the paper, there exists several key points that requires careful justification to make it as a much stronger piece of work. I list them in the below. If these questions could be handled properly, I will increase my score.

1.	In the Remark 1, it seems an very important conclusion the authors want to claim on the role of the second term associated with alpha2. Then what is evidence for the claim that “the choice of alpha2 could affect the tradeoff between accuracy and robustness…alpha2-associated term plays a positive role in boosting robustness, especially for training with large epsilon”?
2.	Why does FAST-BAT produce better gradient alignment? Is it just an empirical observation or is there any good theoretical justification?
3.	Is there any connection between FAST-BAT and standard PGD adversarial training?

Other questions:
About the Hessian-free assumption. What about the non-ReLU activation function for which the function is not piece-wise linear wrt to input?


**Summary Of The Paper:**

This paper aims to interpret the fast adversarial training methods from the perspective bi-level optimization. Though the discovery is straightforward, it is worthy of letting the community know this important connection. And then the authors proposed a new linearization of the lower-level optimization problem and introduced a new FAST-BAT approach for improving both accuracy and robustness. Various experiments were conducted to verify the effectiveness of the proposed method.

**Summary Of The Review:**

Good perspective for interpreting fast adversarial training, but in the current version, there exists several key points that requires careful justification to make it as a much stronger piece of work.

---

> ### Author Response · Authors · 2021-11-19
> **Point-to-point response to specific questions (Part II)**
>
> **Q4:** Other questions: About the Hessian-free assumption. What about the non-ReLU activation function for which the function is not piece-wise linear wrt to input?
>
> **A4:** This is another good way for exploration. We choose the smooth Swish[2] and Softplut function as alternatives for the non-smooth ReLU function. We compare the results of using both hessian-presence and hessian-free versions to examine if the hessian-free assumption still holds for the non-ReLU neural network. The results are provided in the table below. As we can see, the use of Hessian does not affect performance much. A similar phenomenon can be observed across different $\epsilon$ and different model activation functions (ReLU, Softplus, and Swish). However, the introduction of Hessian leads to an increase in time consumption by more than 30%. Therefore, the Hessian-free assumption is reasonable across different activation function choices.
>
> |                       Setting                       | SA (%) ($\epsilon=8/255$) | RA-PGD (%) ($\epsilon=8/255$) | SA (%) ($\epsilon=16/255$) | RA-PGD (%) ($\epsilon=16/255$) | Time (s/epoch) |
> | :-------------------------------------------------: | :--------------------: | :------------------------: | :---------------------: | :-------------------------: | :------------: |
> |                    Fast-AT-ReLU                     |         81.88          |           45.44            |         46.13          |           21.75            |       42       |
> | Fast-BAT-ReLU                       (with Hessian)  |         79.54          |           48.52            |          67.93          |            26.12            |      179       |
> | Fast-BAT-ReLU                        (w/o Hessian)  |         79.47          |           48.67            |          67.81          |            26.07            |      135       |
> |                  Fast-AT-Softplus                   |         81.29          |           47.26            |         45.39          |           22.40            |       42       |
> | Fast-BAT-Softplus                   (with Hessian)  |         79.59          |           49.74            |          68.63          |            25.54            |      178       |
> |  Fast-BAT-Softplus                   (w/o Hessian)  |         79.48          |           49.67            |          68.57          |            25.59            |      137       |
> |                    Fast-AT-Swish                    |         75.61          |           44.43            |         52.03          |           23.08            |       49       |
> | Fast-BAT-Swish                      (with Hessian)  |         73.93          |           45.97            |          62.49          |            23.99            |      196       |
> | Fast-BAT-Swish                        (w/o Hessian) |         73.89          |           45.90            |          62.59          |            23.81            |      141       |
>
> *[2] Ramachandran, Prajit and Zoph, Barret and Le, Quoc V. Searching for activation functions, arXiv preprint arXiv:1710.05941, 2017.*

---

> ### Author Response · Authors · 2021-11-19
> **Point-to-point response to specific questions (Part I)**
>
> We thank you very much for acknowledging the importance of our work. We respond to your concerns and feedback below:
>
> **Q1:** What is evidence for the claim that “the choice of alpha2 could affect the tradeoff between accuracy and robustness…alpha2-associated term plays a positive role in boosting robustness, especially for training with large epsilon”?
>
> **A1:** Sorry for the confusion. We should make our statement more precise. The claim is built upon our empirical observations, shown in the following table. We consider the case of robust training with the large $\epsilon$ choice (16/255). As we can see from the table, if $\alpha_2$ is set too small ($\alpha_2 < 0.008\alpha_1$), then both SA and RA will drop significantly. Here $\alpha_1$ is set as the cyclic learning rate and thus not a constant parameter. However, in the $\alpha_2$ interval $ [0.0125\alpha_1, 0.025\alpha_1]$, we observed a tradeoff between standard accuracy (SA) and robust accuracy (RA): That is, the improvement in RA corresponds to a loss in SA. In our experiments, we choose $\alpha_2$ when the tradeoff yields the best RA without suffering a significant drop of SA (which still outperforms the baseline approaches).
>
> | $\alpha_2$ (CIFAR-10, PreActResNet18,$\epsilon=16/255$) | 0.025$\alpha_1$ | 0.0167$\alpha_1$ | 0.0125$\alpha_1$ | 0.008$\alpha_1$ |
> | :-------------------------------------------------: | :--------: | :--------: | :--------: | :--------: |
> |                       SA (%)                        |   **75.06**    |   69.31    |   67.81    |   57.92     |
> |                     RA-PGD (%)                      |   21.42    |   23.34    |   **26.07**    |   20.53    |
>
> **Q2:** Why does FAST-BAT produce better gradient alignment? Is it just an empirical observation or is there any good theoretical justification?
>
> **A2:** This was an empirical observation. However, a more in-depth explanation can be drawn below. We believe that aligned gradients may be just a **necessary condition** for avoiding catastrophic overfitting, but not a sufficient one. This is why Fast-BAT outperforms Fast-AT-GA, which directly penalizes the gradient alignment. A possible justification can be made from the perspective of the flatness of the loss landscape. A higher gradient alignment implies a flatter loss landscape with respect to input perturbations. However, it has been shown that the direct penalization on the norm of the input gradient may not achieve the state-of-the-art model robustness [1].
>
> *[1] Finlay, C. and Oberman, A. M. Scaleable input gradient regularization for adversarial robustness. arXiv preprint arXiv:1905.11468, 2019.*
>
> **Q3:** Is there any connection between FAST-BAT and standard PGD adversarial training?
>
> **A3:** This is a very insightful question. We add some new experiments below to clarify it. The conclusion is that it is difficult to achieve superior performance by applying BLO to PGD adversarial training because the lower-level problem of adversarial training can not be solved in closed form (namely, the lower-level globally optimal solution is not attainable). Therefore, $\boldsymbol\delta^*$ in such a case is just a **suboptimal** solution, which in turn hampers the performance of the bi-level optimization algorithm. To justify the above argument, we implemented the Bi-level Adversarial Training (BAT) algorithm by integrating BLO with the standard AT algorithm. The results of BAT and AT on (CIFAR-10, ResNet-18) are shown in the table below, where the train-time attack budget is $\epsilon = 8/255$. Robust Accuracy (RA) is evaluated under PGD-50-10 with the same $\epsilon$ as for training and results are averaged over 5 independent trials.
>
> | Results | SA($\epsilon=8/255$) | RA($\epsilon=8/255$) |  Time (s)   |
> | :-----: | :------------------: | :------------------: | :---------: |
> |   AT    |        80.16         |        48.22         |  1$\times$  |
> |   BAT   |        79.43         |        48.11         | 3.5$\times$ |
>
> As we can see, with more training time, BAT does not provide any improvement either on RA or SA.

---

> > ### Comment · Reviewer_N6su · 2021-11-29
> > **Thanks a for the response.**
> >
> > Dear authors,
> >
> > Thank you for providing the rebuttal regarding my concerns.
> >
> > After reading the rebuttal, I think that only using empirical observation to justify the balance between the two terms in Eq.(11) is far from convincing.  Since the key point is that if you set alpha2 = 0, the upper-level model parameter updating step reduces to the standard
> > FAST-AT, and thus what role the additional alpha2 term plays really requires an in-depth understanding. At least I expect some intuitive analysis over this term.

---

> > > ### Author Response · Authors · 2021-11-30
> > > **Further response to reviewer's comments**
> > >
> > > Thanks for your follow-up response. The presence of $\alpha_2$ term in Eq. (11) follows a rigorous implicit gradient derivation; see Eq. (3). That is, if one wants to update the gradient of the upper-level objective function, then BLO requests us to have to compute the alpha2 term. This is a theoretical result rather than a heuristic choice. Our empirical experiments were performed to find a better learning rate choice of $\alpha_2$. Such an empirical justification is commonly used for implementation of deep model training. When adjusting $\alpha_2$, we found that this learning rate choice affects the tradeoff between robustness and accuracy, which also aligns with the no free lunch conclusion of a given adversarial training method [1].
> > >
> > > *[1] Tsipras, D., Santurkar, S., Engstrom, L., Turner, A., and Madry, A. Robustness may be at odds with accuracy. In International Conference on Learning Representations, 2019. https://arxiv.org/abs/1805.12152*

---

> ### Author Response · Authors · 2021-11-24
> **Look forward to your post-rebuttal feedback!**
>
> Dear Reviewer N6su,
>
> Thank you very much for taking the time to review our paper. We cherish your comments very much.
> In the original review, you expressed the willingness to turn your rating to the positive side if your questions can be properly handled.  We hope that our posted response including additional experiments as well as clarifications is able to alleviate your concerns. Please refer to our detailed responses in Part I and Part II. Please also feel free to find the [summary of our response to all reviewers](https://openreview.net/forum?id=gzeruP-0J29&noteId=RlLhb9MpRK) and [highlighted general response](https://openreview.net/forum?id=gzeruP-0J29&noteId=y4L0EA0nnP).
>
>
> If you have additional comments, please feel free to let us know. We will make our best effort to address them.
>
> Thank you very much

---

### Official Review · Reviewer_c58a · 2021-11-02

**Correctness:** 3
**Technical Novelty And Significance:** 2
**Empirical Novelty And Significance:** 2
**Recommendation:** 5
**Confidence:** 4

**Main Review:**

Rewriting (robust) AT as a bi-level optimization problem is a nice formalization that allows the use of implicit differentiation for training robust models. This contribution seems however straightforward and the deployed techniques for optimizing the problem (linearized proximal form, differentiable projection) are standard in the literature.
The other advantage of rewriting the problem in this way is in making a distinction between training loss $\ell_{tr}$ and attack loss $\ell_{atk}$, but this does not seem to be explored in the experiments.

The paper presents some inaccuracies when reporting the related work and in the derivations:

Contrary to what stated in the paper, Adversarial Training was proposed before [Madry 2018] in the seminal work of [1]. In [Madry 2018] adversarial training was formulated as a Robust Optimization problem. The attribution should be rectified.

To write Problems (5) and (7) it is never explained how the term $\frac{\lambda}{2}$ squared norm 2 of $\delta - z$ appears. It does not come directly from the linearization otherwise this term would be multiplied by the second order derivate of $\ell_{atk}$ and there wouldn't be any $\lambda$ hyper-parameter. Instead it seems that the $\lambda$ proximal operator is applied, but this is never mentioned.
The assumptions on the loss functions needed in order to linearize the BLO problem are not reported.

In order to compute the implicit gradient of FAST-AT, $\mathcal{P}_{\mathcal{C}}$ does not need to be differentiable as the derivative of the sign function is 0 null on the dimensions that need an update. The statement "the projection function is not smooth, and thus the use of chain rule is not legitimate" is somehow incorrect, as this projection is still differentiable as it is shown later in page 6, it just gives a piece-wise derivative depending on the box constraints.

In Theorem 1, it is not explained why the so-called Hessian-free assumption is needed. Moreover, its naming seems inaccurate as not all the components of the Hessian should be null (in particular those w.r.t $\theta \delta$).

The following notions essential for understanding the paper are not defined: robustness catastrophic overfitting, random corner linearization, gradient alignment.

The methods used as baselines in the experiments are not sufficiently described, namely FAST-AT-GA and PGD-2-AT. Also AutoAttacks should be described and the reason why this type of attack is chosen to measure robust accuracy should be provided. In the tables, the standard deviations should be used to determine which score differences are significant. At the moment, it seems that the result marked in bold is the one with the best average score, but a significance test should be used instead.
Why does PGD-2-AT generally achieve better results than FAST-AT and FAST-AT-GA and comparable with FAST-BAT?

(Minor) The following reference is defined twice: Aleksander Madry, Aleksandar Makelov, Ludwig Schmidt, Dimitris Tsipras, and Adrian Vladu. Towards deep learning models resistant to adversarial attacks. In International Conference on Learning Representations, 2018b.

[1] Christian Szegedy, Wojciech Zaremba, Ilya Sutskever, Joan Bruna, Dumitru Erhan, Ian J. Goodfellow, and Rob Fergus. Intriguing properties of neural networks, ICLR 2014

**Summary Of The Paper:**

The paper proposes a bi-level optimization problem for training models robust to norm constrained adversarial attacks that can be optimized via argmin/implicit differentiation by linearizing the inner problem (the search for the worst perturbation). The new technique is then compared empirically to other existing approximations of the robust training problem (referred to in the paper as Adversarial Training) and shown to provide improved or comparable results in terms of standard and robust accuracy, for different attacks.

**Summary Of The Review:**

I'd tend to reject the paper because of the highlighted inaccuracies and unclear points, and because the contributions seem straightforward applications of BLO theory to adversarial training. Additionally, the new linearization does not seem to significantly improve state-of-the-art results.

## UPDATE
My final evaluation is based on the current version of the paper, as the authors had the chance to update it.
I increased my score to 5, as the inaccuracies have been corrected, namely:
- the attribution of adversarial training has been rectified
- the additional $\lambda$ norm_2 term in the linearized inner problems has been explained
- the phrasing for the Hessian-free assumption and projection's derivative has been corrected

I still tend to reject the paper for the following reasons:
- definition of gradient alignment and autoattacks is not provided
- justification for using autoattacks is not provided. "everybody is doing it" is not a sufficient justification per se
- comparison wrt baselines is based on mean values, hence the reader cannot judge the significance of the improvements. Doubts about the significance of the empirical results has been raised by other reviewers as well.

Finally, I still have some doubts about the contributions of the paper to the BLO literature. In the rebuttal the authors claim "(1) we show that the choice of lower-level linearization (gradient sign-based vs. non-sign case) could be a key to simplifying BLO with lower-level constraints, and (2) we derive the closed-form of the implicit gradient of lower-level constrained BLO assisted by lower-level linearization."
Hoewever the two linearization schemes are not novel, neither the derivative of the projection onto a convex set.

---

> ### Author Response · Authors · 2021-11-19
> **Point-by-point response to the other concerns mentioned in comments (Part II)**
>
> **Q5:** The following notions essential for understanding the paper are not defined: robustness catastrophic overfitting, random corner linearization, gradient alignment.
>
> **A5:** We add some more explanation below:
>
> 1. Robust catastrophic overfitting was defined in the second paragraph of page 2. Specifically, it was termed in [3]. It refers to a large drop of robustness when training with strong adversaries. As Figure 1(a) in our paper shows, catastrophic overfitting occurs **more frequently** on FAST-AT. That is when the train-time $\epsilon$ exceeds **12/255 on CIFAR-10**, then robust accuracy drops to zero.
>
> 2. Random corner linearization: The random corner linearization means the components of linearization point $z$ are randomly chosen either from $\epsilon$ or $-\epsilon$. Please refer to the detailed definition in Remark 2 of page 6 and Table A2.
>
> 3. Gradient alignment: It is a robust regularization proposed by [4] to alleviate robust catastrophic overfitting (see page 2). Specifically, it is defined by the cosine similarity between the gradient of training loss *w.r.t.* the original input and the training loss *w.r.t.* a randomly perturbed input: $\mathbb E_{(x, y)\sim \mathcal D, \eta \sim \mathcal U(-\epsilon, \epsilon)}[\text{cos}(\nabla_x\ell(x, y;\theta), \nabla_x\ell(x + \eta, y;\theta))]$. It serves as a performance indicator to measure the appearance of catastrophic overfitting (seen "Gradient Alignment for 'free' " on page 9)
>
> *[3] Eric  Wong,  Leslie  Rice,  and  J.  Zico  Kolter.   Fast is better than free:  Revisiting adversarial training.   International Conference on Learning Representations, 2020*
>
> *[4] Maksym Andriushchenko and Nicolas Flammarion.  Understanding and improving fast adversarial training.NeurIPS, 2020.*
>
> **Q6:** The methods used as baselines in the experiments are not sufficiently described, namely FAST-AT-GA and PGD-2-AT. Also, AutoAttacks should be described and the reason why this type of attack is chosen to measure robust accuracy should be provided.
>
> **A6:** (Page 2) FAST-AT-GA is the regularized Fast-AT using gradient alignment regularization. (Page 7) PGD-2-AT refers to the 2-step PGD based adversarial training.
>
> AutoAttack [5] is an ensemble of white and black-box attacks. It has been regarded as a standard attack benchmark for robust training (https://robustbench.github.io/).
>
> *[5] Francesco Croce and Matthias Hein. Reliable evaluation of adversarial robustness with an ensemble of diverse parameter-free attacks. In International Conference on Machine Learning, pp. 2206–2216. PMLR, 2020.*
>
> **Q7:** Why does PGD-2-AT generally achieve better results than FAST-AT and FAST-AT-GA and comparable with FAST-BAT?
>
> **A7:** As shown in the original GitHub [repo](https://github.com/tml-epfl/understanding-fast-adv-training/blob/master/sh/exps_diff_eps_cifar10.sh) of FAST-AT-GA, the performance of FAST-AT-GA is not always stable due to the difficulty in finding the best GA regularization parameter. PGD-2-AT is indeed a good baseline and shows higher stability than FAST-AT-GA sometimes (Table 3). However, as shown in Table 2, 4, and Figure 1, Fast BAT outperforms PGD-2-AT in most cases, and the significance can be justified by the relative improvement vs. performance variance. In particular, Fig. 1 shows that the improvement becomes more substantial as the train-time attack budget increases.
>
> **Q8:** In the tables, the standard deviations should be used to determine which score differences are significant. At the moment, it seems that the result marked in bold is the one with the best average score, but a significance test should be used instead.
>
> **A8:** Thanks for your suggestion. We agree that a significant test will definitely strengthen the superiority of our method. Based on our current experiment results, we performed t-test on the improvement of Fast-BAT compared to Fast-AT and Fast-AT-GA in Table 1. Each t-test is based on the results of 10 independent trials.
>
> For $\epsilon=8/255$, the p-value between the RA of Fast-BAT and Fast-AT over a null-hypothesis against an improvement of 2.5% is $0.007$ and the p-value between the RA of Fast-BAT and Fast-AT-GA over a null-hypothesis against an improvement of 2% is $0.0683$. For $\epsilon=16/255$, the p-value between the SA of Fast-BAT and Fast-AT-GA over a null-hypothesis against an improvement of 8% is $5.46\times10^{-3}$.
>
> These small p-values are far less than $0.1$ and therefore, the improvements of Fast-BAT reported in Table 1 are indeed significant. Further, we would like to point out the results of the t-tests are consistent with our observations based on the mean and variance values. Thus, the significance can be deduced by the relative performance improvement of the methods vs. their variances.

---

> > ### Author Response · Authors · 2021-11-21
> > **Point-by-point response to the other concerns mentioned in comments (Part III)**
> >
> > **Q9:** (Minor) The following reference is defined twice: Aleksander Madry, Aleksandar Makelov, Ludwig Schmidt, Dimitris Tsipras, and Adrian Vladu. Towards deep learning models resistant to adversarial attacks. In International Conference on Learning Representations, 2018b.
> >
> > **A9:** We thank you for your scrutiny and we have fixed this problem in our revision.

---

> > ### Comment · Reviewer_c58a · 2021-11-29
> > **The paper has not been updated with this information**
> >
> > This new information should be reported in the paper.
> >
> > 1. Robust catastrophic overfitting and random corner linearization were indeed already defined. I apologise for missing them. However, gradient alignment, baselines and autoattack are still not.
> >
> > 2. "the significance can be deduced by the relative performance improvement of the methods vs. their variances." Standard deviations are still not reported in Tables 3 and 5 and not used for determining gap significance in Tables 2 and 4: e.g. Table 2, $\vareps = 16/255$  FAST-BAT is not significantly better (based on mean and std) than PGD-2-AT for $\varesp= 4, 8$ but it's in bold. There are 6 such evidently non-significant gaps (based on mean and std) wrongly highlighted in these tables.

---

> > > ### Author Response · Authors · 2021-11-30
> > > **Further response to reviewer's comments**
> > >
> > > Thanks for your response. We provide a point-to-point response below.
> > >
> > > [However, gradient alignment, baselines, and AutoAttack are still not.]
> > > 1. We mentioned that gradient alignment is a regularization term in the Introduction (Page 2) and provided a detailed explanation on the function of this term on Page 5, Paragraph “Gradient alignment for ‘free’”.  We will add the mathematical definition of gradient alignment in the revised version.
> > > 2. As for the baselines, we introduced our baselines in a way widely accepted by the community (e.g. [1]), please refer to the Paragraph “Baselines” on Page 7 and Appendix D “DETAILED EXPERIMENT SETTINGS” for more details. If the reviewer still finds anywhere unclear, please give us a more specific suggestion and we are very happy to revise it.
> > > 3. AutoAttack is very known to the community as a benchmark to test the robust accuracy and we mentioned that in the paragraph “Evaluation details” on Page 7 as well. In the updated version. we will add more introduction to it.
> > >
> > > *[1] Maksym Andriushchenko and Nicolas Flammarion. Understanding and improving fast adversarial training.NeurIPS, 2020.*
> > >
> > > [Standard deviations are still not reported in Tables 3 and 5] We will also add the standard deviations to Tables 3 and 5. However, we would like to bring the reviewer's attention that those tables were not mentioned in your first-round review. In [response to the significance of results (Q8)](https://openreview.net/forum?id=gzeruP-0J29&noteId=-yZdHvmECHT), we have followed your suggestion to justify the significance of our results via the P-value.
> > >
> > > [$\vareps = 16/255$ FAST-BAT is not significantly better (based on mean and std) than PGD-2-AT for $\vareps = 4, 8$ but it's in bold] We would like to clarify that the bold mark in all of our tables suggests the Top-1 mean value in each column for clearer demonstration. All the results have suggested that the improvement of Fast-BAT is consistent. For example, in the case of comparison to PGD-2-AT for $\vareps = 4, 8$, it is clear that FAST-BAT can yield a substantial improvement over PGD-2-AT under the evaluation of AutoAttack, which is an ensemble attack and even stronger than PGD attack.
> > >
> > > We thank the reviewer for these suggestions and we will further revise our paper based on the suggestions. However, we strongly believe that none of these comments should be the reason for a “Rejection”. We hope the reviewer can re-evaluate our paper based on our further response.

---

> ### Author Response · Authors · 2021-11-19
> **Point-by-point response to the other concerns mentioned in comments (Part I)**
>
> Besides the clarification above, we would like to respond to other comments mentioned in the review.
>
> **Q1:** Contrary to what stated in the paper, Adversarial Training was proposed before [Madry 2018] in the seminal work of [1]. In [Madry 2018] adversarial training was formulated as a Robust Optimization problem. The attribution should be rectified.
>
> **A1:** We sincerely thank you for pointing this out. We have rectified this in our revision. We also would like to remark that as stated in page 1 the adversarial training that we mentioned refers to the one with the min-max robust optimization formulation.
>
> *[1] Christian Szegedy, Wojciech Zaremba, Ilya Sutskever, Joan Bruna, Dumitru Erhan, Ian J. Goodfellow, and Rob Fergus. Intriguing properties of neural networks, ICLR 2014*
>
> **Q2:** To write Problems (5) and (7), it is never explained how the parameter $\lambda$ and the term squared norm appears.
>
> **A2:** We apologize for this confusion. The motivation for the term $\lambda / 2 \|\boldsymbol\delta - \mathbf z\|_2^2$ is not from the proximal operation in the proximal algorithm. The reason for introducing this quadratic residual term is two-fold: First, this term stabilizes the lower level problem via regularization;  Second,  it ensures the strong convexity of the lower-level objective function (which thus leads to a unique minimizer). Such a strongly convex regularization term is also common in the BLO analysis (*e.g.* [2]) to stabilize the alternative optimization convergence of BLO. We will add the clarification for this quadratic term.
>
> *[2] Uri Shaham, Yutaro Yamada, and Sahand Negahban. 2015. Understanding Adversarial Training: Increasing Local Stability of Neural Nets through Robust Optimization. arXiv preprint arXiv:1511.05432 (2015).*
>
> **Q3:** In order to compute the implicit gradient of FAST-AT, $\mathcal P_{\mathcal C}$ does not need to be differentiable as the derivative of the sign function is 0 null on the dimensions that need an update. The statement "the projection function is not smooth, and thus the use of chain rule is not legitimate" is somehow incorrect, as this projection is still differentiable as it is shown later in page 6, it just gives a piece-wise derivative depending on the box constraints.
>
> **A3:** Please refer to our [General Response 2](https://openreview.net/forum?id=gzeruP-0J29&noteId=qUCAsw5DgEr).
>
> **Q4:** In Theorem 1, it is not explained why the so-called Hessian-free assumption is needed. Moreover, its naming seems inaccurate as not all the components of the Hessian should be null.
>
> **A4**: The Hessian-free assumption is needed to simplify the IG computation in Eq. (21) and (22) of Appendix. Yes, a more precise naming should be the Hessian with respect to the input perturbation variable $\boldsymbol \delta$. We will update it.
>
> Besides, we also would like to mention that the ‘Hessian-free assumption’ w.r.t. $\boldsymbol \delta$ is reasonable. In the revision, we have conducted a new experiment to compare the performance of Hessian-involved Fast-BAT with that of Hessian-free Fast-BAT. All the results are averaged over 5 independent trials with different seeds. As we can see, the SA and RA of Hessian-involved Fast-BAT do not change much. However, the extra Hessian calculation heavily slows down Fast-BAT as around 30% more time is needed for each epoch. Thus, this experiment justifies our assumption, that the influence of the Hessian matrix is very limited.
>
> |             Method              | RA-PGD (%) $\epsilon=8/255$ | RA-PGD (%) $\epsilon=16/255$ | RA-AA (%) $\epsilon=8/255$ | RA-AA (%) $\epsilon=16/255$ | SA (%) $\epsilon=8/255$ | SA (%) $\epsilon=16/255$ | Time(s) |
> | :-----------------------------: | :--------------------: | :---------------------: | :-------------------: | :--------------------: | :----------------: | :-----------------: | :-----: |
> |    Fast-BAT Hessian     Free    |         48.67          |          26.07          |         44.86         |         18.18          |       79.47        |        67.81        |   135   |
> | Fast-BAT Hessian       Involved |         48.52          |          26.12          |         44.81         |         18.31          |       79.54        |        67.93        |   179   |

---

> ### Author Response · Authors · 2021-11-19
> **Clarification of some possible mis-understandings**
>
> Thank you very much for the comments. First, we would like to clarify some possible **misunderstandings** about our work based on the comments.
>
> **1. General response 1 (GR1): Clarification on our contributions.**
>
> **Reply:**   Our contributions are **not straightforward.**  Please see our detailed response in **[highlighted general response post](https://openreview.net/forum?id=gzeruP-0J29&noteId=y4L0EA0nnP)**.
>
> In particular,     the reviewer commented that “The other advantage of rewriting the problem in this way is in making a distinction between training loss and attack loss, but this does not seem to be explored in the experiments.” However, both the BLO interpretation of Fast-AT and the proposed Fast-BAT approach explored this advantage since the lower-level linearization leads to an attack objective different from the original training objective.
>
> **2. General response 2 (GR2)** for comment "The projection function is not smooth, and thus the use of chain rule is not legitimate" is somehow incorrect, as this projection is still differentiable as it is shown later in page 6, it just gives a piece-wise derivative depending on the box constraints.
>
> **Reply:** We apologize for any confusion that we introduced, and will make our statements as precise as possible. However, we believe that there might exist some misunderstanding on the smoothness of the projection operation, and the validity of the chain rule.
>
> 2.1. The projection operation is **not** smooth. As the reviewer said, projection onto a box constraint gives a piecewise linear function (namely, a clipping function), which is not smooth (at the boundary points). Thus, the chain rule cannot be strictly applied. This is why we stated that "the projection function is not smooth, and thus the use of chain rule is not legitimate".
>
>    2.2. The **rigorous derivation** of implicit gradient (IG) of the lower-level constrained BLO should use KKT conditions as Theorem 1. On page 6, we stated, “a **coarse understanding** of the IG formula (10) can be drawn from the chain rule”. We made this statement just for the purpose of offering an elementary understanding of Eq. (10). In fact, the direct application of the chain rule to $\frac{d \boldsymbol \delta^*(\boldsymbol \theta)}{d \boldsymbol \theta}$ is not precisely the same as Eq. (10). A key difference lies in $\mathbf H_{\mathcal C}$: If we use the chain rule (which does not rigorously hold for a nonsmooth projection function), then the derivative of the projection operation needs to be evaluated at $\mathbf z - (1/\lambda) \nabla_{\boldsymbol \delta = \mathbf z} \ell_{\mathrm{atk}}$ rather than $\boldsymbol \delta^*$ (that appears in $\mathbf H_{\mathcal C}$).
>
>   2.3. Fast-AT is a special case, where the sign of gradient is used. However, as we stated on page 5 that a more rigorous derivation of IG should be given by KKT conditions similar to Theorem 1. Following the proof of Theorem 1, the presence of sign operation in the lower-level objective function of Eq. (4) (noted by g) will lead to the second-order derivative $\nabla_{\theta, \delta} g = 0$ at the lower-level solution. As a result, based on Eq. (21) in Appendix, we can obtain Eq. (6). The reviewer commented that “In order to compute the implicit gradient of FAST-AT, $\mathcal P_{\mathcal C}$ does not need to be differentiable as the derivative of the sign function is $0$ null on the dimensions that need an update.” This will not lead to rigorous proof, since the smoothness of projection operation will affect the validity of the chain rule as discussed above.

---

> > ### Comment · Reviewer_c58a · 2021-11-29
> > **reply on theoretical results**
> >
> > In general, I find the wording and additional explanations (e.g. on the $\lambda$ norm2 term) in the current version of the paper more precise and easier to understand.
> >
> > To answer to the rebuttal's comments:
> >
> > 1. "it is nontrivial to identify the relationship between the gradient sign-based PGD attack generation and the lower-level linearization of BLO."
> > I agree it is nontrivial, but in the paper this relationship is not identified as it is not shown how the linearization of the inner objective is performed to arrive to the gradient sign-based PGD attack's objective. What the paper proposes is an interpretation, not a rigorous derivation.
> >
> > 2. "Even in the line of optimization research, the work on how to efficiently solve lower-level constrained BLO problems is very limited. Thus, our optimization approach is novel. (Sec. 4)"
> > If this is the case, what are exactly the contributions that the paper brings to the BLO literature?
> >
> > 3. "both the BLO interpretation of Fast-AT and the proposed Fast-BAT approach explored this advantage since the lower-level linearization leads to an attack objective different from the original training objective."
> > This statement is in contradiction with the paper's notation. In Problem (4), for instance, $\ell_{atk}$ refers to the loss function and not the whole objective function. In page 4, when suggesting that $\ell_{atk} \neq \ell_{tr}$ the reader is led to think that the loss functions can be different and not that the whole objective functions are.
> >
> > For the record, I have never questioned the non-smoothness of the projection function, but the phrasing and the novelty of the derivation of its derivative.

---

> > > ### Author Response · Authors · 2021-11-30
> > > **Further response to reviewer's comments**
> > >
> > > Thank you very much for your response. We provide a point-to-point response below.
> > >
> > > [What the paper proposes is an interpretation, not a rigorous derivation.] We respectfully disagree. The interpretation of Fast-AT using BLO with lower-level linearization is not an intuitive explanation, but a rigorous derivation. Please refer to Eq. (4) ~ Eq. (6) in our paper for justification. As shown in Appendix C of the revised paper, KKT conditions can be used to provide the mathematical proof for Eq. (6). Note that our justification of the BLO interpretation of Fast-AT is as rigorous as that of Fast-BAT in Theorem 1. The key difference in the proof is the linearization scheme for the lower-level objective function.
> > >
> > > [What are exactly the contributions that the paper brings to the BLO literature?] Compared to existing BLO literature, (1) we show that the choice of lower-level linearization (gradient sign-based vs. non-sign case) could be a key to simplifying BLO with lower-level constraints, and (2) we derive the closed-form of the implicit gradient of lower-level constrained BLO assisted by lower-level linearization.
> > >
> > > [In page 4, when suggesting that $\ell_{\text{atk} \neq \ell_{\text{lr}}}$ the reader is led to think that the loss functions can be different and not that the whole objective functions are.] We understand the reviewer’s concern and our intention was to refer to the entire lower-level objective function of the bi-level optimization problem. We will make this part clearer. Thanks for your suggestion.
> > >
> > > All in all, we strongly believe that the novelty and contributions of our work are sufficient, either theoretically or empirically. To the best of our knowledge, our attempt to leverage BLO to solve adversarial training problems and adopt the lower-level linearization for an explicit derivation of the BLO solver is new to the community.

---

> ### Author Response · Authors · 2021-11-24
> **Look forward to your post-rebuttal feedback!**
>
> Dear Reviewer c58a:
>
> We appreciate the time and effort you spent on reviewing our paper. The comments and suggestions are valuable. In our posted responses, we have conducted additional experiments and provided detailed clarifications to alleviate your concern about our contributions. Please refer to our clarifications of possible misunderstandings and point-to-point responses. Please also feel free to find a summary of our response to all reviewers at [here](https://openreview.net/forum?id=gzeruP-0J29&noteId=RlLhb9MpRK) and highlighted general response at [here](https://openreview.net/forum?id=gzeruP-0J29&noteId=y4L0EA0nnP).
>
> We hope that you find our effortful response convincing. If you have any further questions, please do not hesitate to contact us and we are very happy to discuss them with you here.
>
> Authors

---

> ### Author Response · Authors · 2021-12-02
> **Thanks for raising the score and follow-up response.**
>
> We thank you very much for re-evaluating our work based on our responses and acknowledging our contributions to adversarial training. Regarding the concern reflected in your latest update, we would like to mention the following:
>
> [Definition of gradient alignment and autoattacks is not provided]
> Thanks, but this can readily be fixed. We will present a more detailed definition of gradient alignment and AutoAttack in the updated version. In the current version, we did not miss any reference on gradient alignment and AutoAttack. And gradient alignment is explained in "Gradient Alignment for 'free' " on page 9. In the adversarial learning community, AutoAttack is a known robust evaluation benchmark, and was often directly cited in the literature, e.g., [1] (Paragraph “Results on CIFAR-10 and SVHN ” on Page 8) and [2] (Sec. 5).
>
> *[1] Maksym Andriushchenko and Nicolas Flammarion. Understanding and improving fast adversarial training.NeurIPS, 2020*
>
> *[2] Fan, L., Liu, S., Chen, P. Y., Zhang, G., & Gan, C. When Does Contrastive Learning Preserve Adversarial Robustness from Pretraining to Finetuning?. NeurIPS, 2021*
>
> [Justification for using autoattacks is not provided. "everybody is doing it" is not a sufficient justification per se]
> Thanks, but the reviewer might misunderstand the reason for using AutoAttack. First, AutoAttack is an ensemble attack, known as the strongest robust benchmark evaluation metric. Thus, this paper, together with the baselines [Fast AT with gradient alignment] used AutoAttack to evaluate adversarial robustness. Second, AutoAttack is easier to compare across different methods using the [leader board](https://robustbench.github.io/). We were not aware of any other attack benchmark more convincing than AutoAttacks.
>
> [Comparison wrt baselines is based on mean values, hence the reader cannot judge the significance of the improvements. Doubts about the significance of the empirical results have been raised by other reviewers as well.]
> We respectfully disagree that "comparison w.r.t. baselines is based on mean values". In main tables 1, 2, and 4, we have included standard deviations. In response to question [Q8] (https://openreview.net/forum?id=gzeruP-0J29&noteId=-yZdHvmECHT), we also conducted significant tests to demonstrate the significance.
>
> **In a nutshell, we highly appreciate Reviewer c58a's follow-up comments. However, we sincerely encourage the reviewer to re-think the listed main reasons for rejecting our work. They can readily be addressed.**
>
> [BLO literature novelties: I still have some doubts about the contributions of the paper to the BLO literature. In the rebuttal the authors claim " (1) we show that the choice of lower-level linearization (gradient sign-based vs. non-sign case) could be a key to simplifying BLO with lower-level constraints, and (2) we derive the closed-form of the implicit gradient of lower-level constrained BLO assisted by lower-level linearization." However the two linearization schemes are not novel, neither the derivative of the projection onto a convex set.]
> The use of sign-based linearization and KKT conditions to **prove** the simplification of implicit gradients is novel. The systematic BLO with lower-level linearization is novel. Maybe the techniques that we used, e.g., linearization, projection, KKT are not sound novel to the reviewer. However, having them as a **coherent, theoretically-grounded optimization solution** to solving adversarial training problems is novel.

---

> ### Author Response · Authors · 2021-12-08
> **Table 3 and Table 5 with standard deviations**
>
> Dear Reviewer c58a:
>
> As you recently commented that standard deviations are missing in Tables 3 and 5, we have added these results; see [revision of Table 3](https://ibb.co/PFRjXwQ) and [revision of Table 5](https://ibb.co/n8Cd66s). All the mean values and standard deviations are computed over 10 independent trials. Through the reported standard deviations (together with our [latest responses](https://openreview.net/forum?id=gzeruP-0J29&noteId=lrN-yFXRTWV)), we hope to convince Reviewer c58a that the improvement of our proposed method is consistent as well as significant, since the standard deviations are much smaller compared to the improvement on mean values. In addition to 'following the standard evaluation pipeline', we have tried our best to make our evaluations as comprehensive as possible.
>
> We would like to kindly highlight that to the best of our knowledge, our work provides a theoretically-grounded optimization framework for fast AT (BLO + sign-based lower-level linearization, proved by KKT-based implicit gradients) for the first time. This is also the first work to improve fast AT using BLO with lower-level linearization. The BLO-based optimization system is quite unique compared to existing adversarial training methods.
>
> Thank you very much,
>
> Authors

---

### Official Review · Reviewer_TZX8 · 2021-11-02

**Correctness:** 3
**Technical Novelty And Significance:** 3
**Empirical Novelty And Significance:** 3
**Recommendation:** 6
**Confidence:** 4

**Main Review:**

Relevance: the paper focuses on the important problem of designing computationally attractive algorithms for adversarial training -- this is an important problem that is relevant to ICLR.

Results: The main strength of the paper is the empirical study. Given that within the same computation budget as the baselines, the proposed method has lower variance and resolves catastrophic overfitting in several settings, I found the results important. The results are only averaged over 5 random experiments, therefore, given the complexity and the size of the task, I’m not sure how reliable the estimates of mean/variance are. I think the empirical results can be much more compelling if the authors include more experiments.

Presentation: Although the paper is well-written for the most part, there is much room for improvement in sections 3 and 4. In particular, some claims are not well-justified, there are some issues with the notation and the presentation lacks mathematical rigor. Here are some examples:

            - in page 3, the authors state that “FAST-AT-GA yields improved robustness but has a poor accuracy-robustness tradeoff (e.g. Table 1)“. I do not see how FAST-AT-GA has a poor accuracy-robustness tradeoff in Table 1. Can you elaborate?

            - the description of set C = {delta | || delta ||_infty < epsilon,  delta in [0, 1]} is very confusing. What does delta in [0, 1] mean? attacks are always point-wise positive? How is it not vacuous since epsilon < 1?

            - in page 4, the authors say “Even if we set ell_atk = - ell_tr, problem 2 does not reduce to 1 due to the presence of lower-level constraint”. Regardless of whether this claim is true or not, the justification is far from satisfactory. In order to argue two optimization “problem” are equivalent, or if one reduces to another, one needs to prove/disprove that solutions to one constitute solutions to the other. The fact that gradients are different — as argued in the paper — does not a-priori inform us of anything useful about the equivalence of the two. In particular, when ell_atk = - ell_tr, then delta* in (2) is very well a maximizer of the inner attack problem in (1). Again, whether two optimization problems are equivalent has nothing to do with certain optimization procedures (such as PGD being considered here) generating different trajectories.

            - the bi-level interpretation of Fast-At is trivial — this is how gradient updates are motivated in the literature of first-order optimization.

            - in page 5, the derivation of the implicit gradient in (6) seems reasonable but lacks mathematical rigor. I suggest authors include careful proof here.

            - The Hessian-free assumption in Theorem 1 essentially means that the attack loss ell_atk is linear. I do believe that this is not true for most interesting cases, and I do not buy the argument “the rationale behind the Hessian-free assumption is that neural networks commonly lead to a piece-wise linear decision boundary”. Are you suggesting that the loss landscape in deep learning is almost linear?

            - in page 6, remark 1, the authors state that “Clearly, if alpha_2 = 0, then it reduces to the standard Fast-AT”. How is this true? here the updates will be based on the gradient of the loss whereas in standard Fast-At the updates are based on the sign of the gradient.


**Summary Of The Paper:**

The paper studies adversarial training as a bi-level optimization problem. The authors show that Fast Adversarial Training can be viewed as a bi-level optimization problem where the lower-level problem is a linearization of the loss in the direction of the sign of the gradient. Motivated by this observation, the authors propose Fast Bi-level Adversarial Training, where the lower-level optimization linearizes the loss in the direction of the gradient (as opposed to the sign of the gradient). The authors then show analytically how to obtain the gradient of the upper-level problem under the assumption that the loss Hessian is equal to zero. This gradient is then used in an iterative manner similar to standard adversarial training. The proposed method is evaluated on CIFAR10 and ImageNet against multiple baselines including Fast-AT, Fast-AT-GA, and PGD-2. The empirical results show that compared to the baselines, within the same order of computational cost, the proposed method enjoys improved stability and mitigates the catastrophic overfitting present in other baselines.

**Summary Of The Review:**

To summarize:
Strengths:
          - the paper studies an important and relevant problem.
          - the experiments are well done.
          - the proposed adversarial training approach has nice theoretical motivations.
          - the paper is well-written for the most part.

Weakness:
          - several ideas and claims are handwavy and lack mathematical rigor
          - the assumption behind the main theoretical result is too stringent
          - statistical significance --> only 5 random experiments

---

> ### Author Response · Authors · 2021-11-19
> **Point-to-point response to specific questions (Part III)**
>
> **Q6:** In page 5, the derivation of the implicit gradient in (6) seems reasonable but lacks mathematical rigor. I suggest authors include careful proof here.
>
> **A6:** Thanks for this suggestion and we will include formal proof.  As we stated after Eq. 6, the proof will be similar to that of Theorem 1 using KKT conditions to derive the implicit gradient. The key difference is that the presence of sign operation in the lower-level objective function of Eq. 4 (noted by $g$) will lead to the second-order derivative $\nabla_{\theta, \delta} g = 0$ at the lower-level solution. As a result, based on Eq. 21 in Appendix, we can obtain Eq. 6.
>
> **Q7:** The Hessian-free assumption in Theorem 1 essentially means that the attack loss $\ell_{atk}$ is linear. I do believe that this is not true for most interesting cases, and I do not buy the argument “the rationale behind the Hessian-free assumption is that neural networks commonly lead to a piecewise linear decision boundary”. Are you suggesting that the loss landscape in deep learning is almost linear?
>
> **A7:** Thanks for your question. First, there was a typo in the Hessian-free assumption of Theorem 1. It should apply to the input perturbation variable $\boldsymbol \delta$ rather than $\boldsymbol \theta$. We apologize for the confusion. Second, we would like to respectfully point out that the loss landscape of the ReLU-based deep neural networks is indeed piece-wise linear or polyhedral (which is not a simple linear function); see [5] and [6] for more information regarding this claim. In fact, this argument is also acknowledged by another reviewer N6su in his/her [comments](https://openreview.net/forum?id=gzeruP-0J29&noteId=a3laIqJRpdr) (paragraph "other question"). Third, to examine how the Hessian $\nabla_{\boldsymbol \delta \boldsymbol \delta} \ell_{atk}$ affects the performance of Fast-BAT, we add a **new experiment below** to compare the Hessian-free Fast-BAT with the Hessian-aware Fast-BAT. As we can see,  there is not much performance difference when Hessian is used. However, the extra Hessian calculation heavily slows down Fast-BAT as around 30% more time is needed for each epoch.
>
> |             Method              | RA-PGD (%) $\epsilon=8/255$ | RA-PGD (%) $\epsilon=16/255$ | RA-AA (%) $\epsilon=8/255$ | RA-AA (%) $\epsilon=16/255$ | SA (%) $\epsilon=8/255$ | SA (%) $\epsilon=16/255$ | Time(s) |
> | :-------------------: | :--------------------: | :---------------------: | :-------------------: | :--------------------: | :----------------: | :-----------------: | :-----: |
> | Fast-BAT w/o Hessian  |         48.67          |          26.07          |         44.86         |         18.18          |       79.47        |        67.81        |   135   |
> | Fast-BAT with Hessian |         48.52          |          26.12          |         44.81         |         18.31          |       79.54        |        67.93        |   179   |
>
>
> *[5] Seyed-Mohsen Moosavi-Dezfooli, Alhussein Fawzi, Jonathan Uesato, and Pascal Frossard. Robustness via curvature regularization, and vice versa. In Proceedings of the IEEE Conference on Computer Vision and Pattern Recognition, pp. 9078–9086, 2019.*
>
> *[6] Motasem Alfarra, Adel Bibi, Hasan Hammoud, Mohamed Gaafar, and Bernard Ghanem. On the decision boundaries of deep neural networks: A tropical geometry perspective. arXiv preprint arXiv:2002.08838, 2020.*
>
>
> **Q8:** In page 6, remark 1, the authors state that “Clearly, if alpha_2 = 0, then it reduces to the standard Fast-AT”. How is this true? Here the updates will be based on the gradient of the loss whereas in standard Fast-At the updates are based on the sign of the gradient.
>
> **A8:** We believe there might exist a misunderstanding regarding Remark 1. Eq. (11) corresponds to the **model parameter updating step** given the attack instance $\boldsymbol \delta^*$. Note that in Fast-AT, the sign of gradient is only used for generating $\boldsymbol \delta^*$ rather than updating model parameters. In fact, to update model parameters, both Fast-AT and Fast-BAT utilize stochastic gradient descent (SGD). However, the gradient of upper-level model training in Fast-BAT involves the implicit gradient, corresponding to the $\alpha_2$-associated term. Thus, we meant if $\alpha_2 = 0$, then the **upper-level model training** of Fast-BAT, i.e., Eq. (11), will reduce to the model training step of Fast-AT. If our previous statement causes confusion, we apologize for that and will make it more precise.

---

> ### Author Response · Authors · 2021-11-19
> **Point-to-point response to specific questions (Part II)**
>
> **Q4:** In page 4, the authors say “Even if we set $\ell_{atk} = - \ell_{tr}$, problem 2 does not reduce to 1 due to the presence of lower-level constraint”. Regardless of whether this claim is true or not, the justification is far from satisfactory.
>
> **A4:** Thank you very much for the comment. We agree that some explanation should be added for ease of understanding from the BLO perspective. In what follows, we will justify our claim from the following two points.
>
> 1. In the **absence** of the constraint $\boldsymbol \delta \in \mathcal C$, if we set $\ell_{\text{atk}} = -\ell_{\text{tr}}$, then problem (2) will reduce to (1).
>
>    This is a known BLO result (*e.g.* [3]) and can be readily proven using the stationary condition. To be specific, based on the stationary condition of unconstrained lower-level optimization, we have $\nabla_{\boldsymbol \delta}\ell_{\text{atk}}(\boldsymbol\theta, \boldsymbol\delta^*) = 0$. Since $\ell_{\text{atk}} = -\ell_{\text{tr}}$, we have $\nabla_{\boldsymbol \delta}\ell_{\text{tr}}(\boldsymbol\theta, \boldsymbol\delta^*) = 0$. As a result, the second term in Eq. (3) becomes $\mathbf 0$ and solving problem (2) becomes identical to solving the min-max problem (1).
>
> 2. In the **presence** of the constraint $\boldsymbol \delta \in \mathcal C$, the stationary condition cannot be applied since the stationary point may not be a feasible point in the constraint. In other words, $\nabla_{\boldsymbol \delta}\ell_{\text{atk}}(\boldsymbol\theta, \boldsymbol\delta^*) = 0$ does not hold in the case of $\ell_{\text{atk}} = -\ell_{\text{tr}}$. As a matter of fact, one has to resort to KKT conditions instead of the stationary condition for a constrained lower-level problem. Similar to our proof in Theorem 1, the implicit gradient (and thus the second term of Eq. (3)) cannot be omitted in general. This makes problem (2) different from the problem (1).
>
> *[3] Ghadimi, S. and Wang, M. Approximation methods for bilevel programming. arXiv preprint arXiv:1802.02246, 2018.*
>
> **Q5:** The bi-level interpretation of Fast-At is trivial — this is how gradient updates are motivated in the literature of first-order optimization.
>
> **A5:** We respectfully disagree.
>
> First, the interpretation of Fast-AT through the lower-level linearization in BLO might be direct once we expressed it out. However, this is non-trivial. We are not aware of any study on robust training through the lens of BLO. Compared to min-max optimization, BLO allows the (lower-level) attack objective to be made different from the (upper-level) training objective. This novel insight allows us to shape Fast-AT using a different (linearized) attack objective in the BLO context.
> It is our contribution to unveiling the connection between the one-step sign-based PGD attack and the lower-level linearization of BLO. Furthermore, this BLO view enables us to improve Fast-AT using a different linearization scheme. Even in the optimization community, we are not aware of any work that analyzes the BLO problem with lower-level linearization. The reviewer is very welcome to list any literature that we might have missed.
>
> Second, in the literature on adversarial attack and adversarial defense, we are only aware of the existing interpretation between the fast gradient sign method (FGSM) based attack and linearized first-order optimization [4]. However, in Fast-AT, **the attack generation is done by one-step PGD attack rather than FGSM** since the former needs the projection operation so as to use the learning rate greater than the attack budget $\epsilon$. To the best of our knowledge, no work has interpreted PGD-based Fast-AT as we did.
>
> *[4] Uri Shaham, Yutaro Yamada, and Sahand Negahban. 2015. Understanding Adversarial Training: Increasing Local Stability of Neural Nets through Robust Optimization. arXiv preprint arXiv:1511.05432 (2015).*

---

> ### Author Response · Authors · 2021-11-19
> **Point-to-point response to specific questions (Part I)**
>
> We sincerely thank your acknowledgment of the significance of our work and the importance of our empirical results. In what follows, we will provide the point-to-point response to the concerns mentioned in the comments.
>
> **General Response:** Presentation issue of Sec. 3 and 4.
>
> Thank you very much for the very careful review. We apologize for our typos and possibly vague statements. However, we would like to mention that we intended not to shape Sec. 3 and Sec. 4 in a math-involved fashion for ease of understanding. Our claims are mathematically rigorous, as clarified by our responses below. Even in the initial submission, we have made a significant effort to improve the readability of our technical sections. We will further improve them based on the reviewer's suggestion.
>
>
>
> **Q1:** The results are only averaged over 5 random experiments, therefore, given the complexity and the size of the task, I’m not sure how reliable the estimates of mean/variance are. I think the empirical results can be much more compelling if the authors include more experiments.
>
> **A1:** We thank you for your suggestion. In fact, we strictly followed the multi-trial setup widely accepted by the community. FAST-AT-GA [1] averaged their main results over 5 independent trials and FAST-AT [2] over only 3. Nevertheless, we will still take your suggestion and conduct five more trials for each setting so that each result is supported by 10 independent trials in total (which take more computation time). The **new results** are presented [here](https://ibb.co/0ZbvdnT) and have been added to the revision of our paper.  As you can see, the additional 5 experiments do not make a great difference to the previously reported mean and variance value in Table 1 in our paper (see a screenshot [here](https://ibb.co/6BLJtgc) for comparison). Therefore, we believe our previous results are reliable.
>
> *[1] Maksym Andriushchenko and Nicolas Flammarion. Understanding and improving fast adversarial training. NeurIPS, 2020.*
>
> *[2] Eric Wong, Leslie Rice, and J. Zico Kolter. Fast is better than free: Revisiting adversarial training. In International Conference on Learning Representations, 2020.*
>
> **Q2:** In page 3, the authors state that “FAST-AT-GA yields improved robustness but have a poor accuracy-robustness tradeoff (e.g. Table 1)“. I do not see how FAST-AT-GA has a poor accuracy-robustness tradeoff in Table 1. Can you elaborate?
>
> **A2:** If we compare Fast-BAT with Fast-AT-GA along with the columns RA-PGD (%) ($\epsilon$ = 16/255) and SA (%) ($\epsilon$ = 16/255), then we can observe that although Fast-AT-GA yields a similar RA to Fast-BAT, i.e., 26.29% vs. 26.07%, the former yields a much poorer SA than the latter, i.e., 58.53% vs. 67.81%. Thus, FAST-AT-GA achieves a good RA at the significant cost of SA degradation, leading to a poorer accuracy-robustness tradeoff.
>
> **Q3:** The description of set $C = \{\delta | || \delta ||_\infty < \epsilon, \delta \in [0, 1]\}$ is very confusing. What does $ \delta \in [0, 1]$ mean?
>
> **A3:** Thank you very much for the careful review. There exists a typo in the description of the set $C$. We are supposed to have the following constraints $\delta + x \in [0, 1]$ instead of $\delta \in [0, 1]$. Therefore, $\delta$ can either be positive or negative. In spite of this typo, our presentation of the linear constraint in Eq. (9) is correct, as the definition of $\mathbf b$ was indeed derived following $C = \{\delta | || \delta ||_\infty < \epsilon, \delta + x \in [0, 1] ] \in [0, 1]\}$.

---

> ### Author Response · Authors · 2021-11-24
> **Look forward to your post-rebuttal feedback!**
>
> Dear Reviewer TZX8:
>
> Thank you very much for your precious comments and suggestions for our paper. In our earlier posted responses, we conducted new experiments to empirically justify our “Hessian-free assumption” and provided additional clarifications including some rigorous proof to alleviate your concerns. Please refer to our point-to-point response listed in Part [I-III] respectively. Please also feel free to find a summary of our response to all reviewers at [here](https://openreview.net/forum?id=gzeruP-0J29&noteId=RlLhb9MpRK) and highlighted general response at [here](https://openreview.net/forum?id=gzeruP-0J29&noteId=y4L0EA0nnP).
>
> We hope that you find our effortful response convincing. If you have additional concerns or comments, please feel free to let us know. We will make our best effort to address them.
>
> Authors

---

### Official Review · Reviewer_6Qs9 · 2021-11-04

**Correctness:** 2
**Technical Novelty And Significance:** 3
**Empirical Novelty And Significance:** 2
**Recommendation:** 3
**Confidence:** 4

**Main Review:**

Strengths: the work outperforms Fast-AT-GA at $\epsilon=16/255$, obtaining better standard accuracy in less time while matching robust accuracy (standard accuracy ~68% instead of ~59%).

Weaknesses:
Baselines and experimental evaluation:
- When comparing performance on a system that has evaluation metrics of both accuracy and training speed, the evaluation must take both into account. Currently it does not: all we see is a table with times and accuracies. One standard way of comparing training speed and accuracy is to measure either the "maximum accuracy in a fixed amount of time" or "minimum time to attain a fixed accuracy" while varying the metric of interest (time or accuracy respectively). Using this type of evaluation, a reader can see the speed/adv. accuracy frontier necessary to compare across algorithms. Currently the evaluation is not sufficiently detailed to compare across algorithms.
- As a suggestion, one natural way of trading off time for accuracy is to treat the number of epochs as a free parameter; an illuminating here would be adv. performance vs training time as the number of epochs changes for each method.
- The performance analysis is currently not informative; it uses a fixed training configuration for each training method (according to the Table 1 description, all the algorithms use early stopping yet according to Figure 1 this heavily disadvantages Fast-AT compared to no early stopping). One must choose the pareto (trading off speed and accuracies) optimal configurations for each training algorithm. The analysis must ablate across different training configurations to be fair.
- The time does not have a standard deviation.
- How is it that $\epsilon=8/255$ and $\epsilon=16/255$ trained models always have the same runtime, as reported in Table 1? Does early stopping always exit the same epoch for both?

Characterization of previous work:
- What is the difference between problem (i) and problem (ii) on page 2? Robust performance is always properly evaluated against the strongest possible adversary.
- The authors claim to solve an issue with Fast-AT stemming from its tendency towards catastrophic overfitting. The authors should give more background on when Fast-AT fails to catastrophic overfitting; how often does it happen? How does the $\epsilon$ impact Fast-AT's convergence?

**Summary Of The Paper:**

This work focuses on the problem of speeding up adversarial training in the $\ell_\inf$ threat model. The work first describes two previous works designed to speed up adversarial training:
- Fast-AT: carefully perform single-step PGD (otherwise known as FGSM) to train $\ell_\inf$ robust models - from Wong, Rice and Kolter 2020
- Fast-AT-GA: single-step PGD + a gradient alignment loss function - from Andriushchenko and Flammarion 2020.

The work then lists a problem with Fast-AT:
- Fast-AT experiences catastrophic overfitting (i.e. provides no robustness on held-out data) for $\epsilon = 16/255$ (Note: Fast-AT-GA does not have this catastrophic overfitting issue)

Finally, the authors present a new method based on bi-level optimization to perform fast $\ell_\inf$-robust learning. They evaluate it and compare with Fast-AT-GA using a 20 epoch schedule, finding that for $\epsilon = 16/255$, the technique can provide the same robustness except with standard accuracy ~68% instead of ~59%.

**Summary Of The Review:**

While the work shows a promising approach (as measured by outperforming Fast-AT-GA at $\epsilon=16/255$ in terms of standard accuracy while beating it in speed and in robust accuracy), the evaluation is lacking. In particular, the evaluation does not give a speed/accuracy tradeoff for each training routine, and does not properly hyperparameter search for each compared algorithm in the comparison. Finally, the work has some minor issues in its characterization of previous work.

---

> ### Author Response · Authors · 2021-11-19
> **Point-to-point response to specific questions (Part II).**
>
> **Q4:** The time does not have a standard deviation. How is it that ϵ=8/255 and ϵ=16/255 trained models always have the same runtime, as reported in Table 1? Does early stopping always exit the same epoch for both?
>
> **A4:** Thanks for your suggestion. We did not report the standard deviation of time as it is very small, [variance < 0.01 ] for both ϵ=8/255 and ϵ=16/255. This is not surprising since the choice of ϵ=8/255 and ϵ=16/255 will only affect the clipping operation (used to project the perturbation onto $\ell_\infty$ balls of different $\epsilon$ radius), and the number of attack generation steps and model updating steps remain intact. Thereby, the runtime does not change much given a method.
>
> **Q5:** What is the difference between problem (i) and problem (ii) on page 2?
>
> **A5:** Problem (i) reflects the instability of existing robust training algorithms trained and evaluated for a fixed perturbation strength $\epsilon$. This is a static single-$\epsilon$ evaluation. That is, with a fixed $\epsilon$, we monitor the variance of performance achieved by each method over repetitive experiments. For example, we have clarified this issue on page 2 when presenting the results of Table 1: RA-PGD using Fast-AT and Fast-AT-GA yields a larger variance than that of using our approach at $\epsilon = 8/255$.
>
> By contrast, problem (ii) refers to catastrophic robust overfitting, which is measured by tracking the robust accuracy over time along with the increase of the train-time perturbation strength $\epsilon$. Thus, this is a dynamic multi--$\epsilon$ evaluation. For example, we have clarified this issue on page 2 when presenting results of Table 1: When a stronger train-time attack (*i.e.*, $\epsilon = 16/255$) is adopted, FAST-AT suffers a sharp drop in RA compared to the robust training of using $\epsilon = 8/255$. This can also be observed in Fig 1a (the orange line).
>
> **Q6:** The authors should give more background on when Fast-AT fails to catastrophic overfitting; how often does it happen? How does the $\epsilon$ impact Fast-AT's convergence?
>
> **A6:** Thanks for your comment. We will add more background and discussions to the paper. Robust catastrophic overfitting was termed in [1]. It refers to a sharp drop of robustness when training with strong adversaries. As Figure 1(a) in our paper shows, catastrophic overfitting occurs **more frequently** on FAST-AT. That is when the train-time $\epsilon$ exceeds **12/255 on CIFAR-10**, then robust accuracy drops to zero almost for sure.
>
> *[1] Maksym Andriushchenko and Nicolas Flammarion. Understanding and improving fast adversarial training. NeurIPS, 2020.*

---

> ### Author Response · Authors · 2021-11-19
> **Point-to-point response to specific questions (Part I).**
>
> **Q1:** One standard way of comparing training speed and accuracy is to measure either the "maximum accuracy in a fixed amount of time" or "minimum time to attain a fixed accuracy" while varying the metric of interest (time or accuracy respectively).
>
> **A1:** The reviewer casts doubts on our evaluation metrics on the training speed-accuracy comparison and provides two ways of comparing training speed and accuracy, "maximum accuracy in a fixed amount of time" or "minimum time to attain a fixed accuracy". We thank you very much for your suggestion. Please see our response below.
>
> 1.  Our evaluation setup is commonly used, e.g., in Fast-AT (Wong et al., 2020). We measure the maximum **robust** accuracy in a fixed number of training epochs and then report the achieved accuracies (including standard accuracy, and robust accuracies against PGD attacks and AutoAttacks) and computation time per epoch.
>
> 2.  Thank you for the suggestion on "maximum accuracy in a fixed amount of time". However, this might not be a direct approach. The reasons are listed below.
>
> 	 - First, if the fixed amount of training time is small, then different robust training methods may have not converged and the resulting robust accuracies may give us a ‘false’ sense of robustness that a method could eventually achieve in practice.
> 	 - Second, if the fixed amount of training time is large, then all methods will suffer from the robust overfitting issue, and thus, the best model is achieved at the earlier training epoch. In this sense, training longer turns out not to be very useful.
>
> 	In spite of the above reasons, we **still follow the reviewer’s suggestion** to add a **new experiment** that reports the maximum standard accuracy (SA) and robust accuracy (RA) versus the amount of training time by treating the epoch number as a free parameter.  The **new results** are presented [here](https://ibb.co/k3RykXD) for $\epsilon=8/255$ and [here](https://ibb.co/wY38G1G) for $\epsilon=16/255$. As we can see, our method converges to the best robust accuracy. Fast-AT is faster than ours at the early training stage due to less computational time for each epoch, but it converges to a sub-optimal solution with much worse robust accuracy.
>
> 3. Thank you for the suggestion on "minimum time to attain a fixed accuracy". However, we do not believe that this is a feasible choice given that all methods need to attain a **fixed** accuracy. In particular, adversarial training is a non-convex min-max optimization problem, it is truly difficult to make baseline methods and ours to achieve the fixed accuracy since they adopted different training configurations and optimization methods (see above [example](https://ibb.co/k3RykXD)).
>
> **Q2:** As a suggestion, one natural way of trading off time for accuracy is to treat the number of epochs as a free parameter; an illuminating here would be adv. performance vs training time as the number of epochs changes for each method.
>
> **A2:** Please see our response to A1-2.
>
> **Q3:** It uses a fixed training configuration for each training method (according to the Table 1 description, all the algorithms use early stopping yet according to Figure 1 this heavily disadvantages Fast-AT compared to no early stopping)
>
> **A3:** As we have clarified in the [**GR2: Clarification on experimental evaluation**], we used the best-reported configuration for each method. And the use of early stopping benefits each method due to the issue of robust overfitting. This is a standard robust training setup.

---

> ### Author Response · Authors · 2021-11-19
> **Clarification of some possible misunderstandings**
>
> We really appreciate your valuable comments. However, we would like to first clarify some possible **misunderstandings** about our work based on the comments.
>
> **->  General response 1 (GR1): Clarification on our contributions.**
>
>    Our contributions go much **beyond** proposing a method for speeding up adversarial training. In light of the contributions listed at the beginning post of [highlighted general response](https://openreview.net/forum?id=gzeruP-0J29&noteId=y4L0EA0nnP) , we kindly hope to bring Reviewer #6Qs9’s attention to the novelties of our work.
>
> **-> General response 2 (GR2): Clarification on experimental evaluation.**
>
>    We **respectfully disagree** that our baselines and experimental evaluations are weak. We believe that all of our experiments follow the baseline repos and the standard robust evaluation pipeline. We would like to make the following clarifications prior to answering your specific questions.
>
>  1. **[Evaluation follows the literature]** Our evaluation metrics and evaluation methods are standard in the adversarial robustness literature, e.g., AT, Fast-AT, and FAST-AT-GA. That is, all methods report standard and robust accuracies, and computation time by using a fixed training epoch budget and taking into account early-stopping to alleviate robust overfitting.
>
>    2.  **[Robust overfitting is a known phenomenon in robust training]** Different from standard training, robust overfitting [1] turns out to be a dominant phenomenon in the adversarially robust training of deep networks. That is, after a certain point of time in robust training, e.g., immediately after the first learning rate decay, the robust test errors will only continue to substantially increase with further training; see Figure 1 of [1] for a visualization. Therefore, nearly all recent methods adopt the early-stopping policy [1] to avoid overfitting or pick the best robust model after the entire training process.
>
>    3.   **[Fair comparison with baselines]** In the paper (as stated on page 7), all baselines are implemented using the recommended training configurations in their official GitHub repos: [Fast-AT](https://github.com/locuslab/fast_adversarial), [Fast-AT-GA](https://github.com/tml-epfl/understanding-fast-adv-training/blob/master/sh/exps_diff_eps_cifar10.sh). We did not disadvantage Fast-AT and other baselines.
>
>    All in all, we hope our general response clarifies some possible misunderstandings that the reviewer may have. In what follows,  we will provide a point-to-point response to specific comments.
>
>    *[1] Leslie Rice, Eric Wong, and Zico Kolter.  Overfitting in adversarially robust deep learning.  InInternationalConference on Machine Learning, pp. 8093–8104. PMLR, 2020.(https://arxiv.org/pdf/2002.11569.pdf)*

---

> ### Author Response · Authors · 2021-11-24
> **Look forward to your post-rebuttal feedback!**
>
> Dear Reviewer 6Qs9,
>
> Thank you very much for taking the time to review our paper. We cherish your comments very much. In our earlier posted responses, we have conducted new experiments regarding your suggestion on the speed-accuracy tradeoff and added additional clarifications to alleviate your concerns about our evaluation plans. Please refer to our response listed in the [clarification of possible misunderstandings](https://openreview.net/forum?id=gzeruP-0J29&noteId=dZTezf6hoeP). Please also feel free to find a summary of our response to all reviewers at [here](https://openreview.net/forum?id=gzeruP-0J29&noteId=RlLhb9MpRK) and highlighted general response at [here](https://openreview.net/forum?id=gzeruP-0J29&noteId=y4L0EA0nnP).
>
> We hope that you find our effortful response convincing. If you have additional comments, please feel free to let us know. We will make our best effort to address them.
>
> Authors

---

> > ### Comment · Reviewer_6Qs9 · 2021-11-28
> > **Response**
> >
> > Thank you for the response. Assuming that all else is held equal (i.e. that the authors do not perform any new systems optimizations that previous work did not) the proposed algorithm empirically matches the speed of Fast-AT-GA.

---

> > > ### Author Response · Authors · 2021-11-28
> > > **Further response**
> > >
> > > Thank you very much for the follow-up comment.
> > >
> > > [Empirically matching to the speed of Fast-AT-GA?] As shown in Table 1, Fast-GA is 10% faster than Fast-AT-GA (135s/epoch vs. 150s/epoch). We would also respectfully point out that as clarified in [Highlighted General Response](https://openreview.net/forum?id=gzeruP-0J29&noteId=y4L0EA0nnP), our contributions are much beyond speeding up adversarial training. For example, in Table 1, our proposed method achieves the best robustness at medium attack strength (e.g. at $\epsilon=8/255$ ) and SOTA robustness at large attack strength (e.g. $\epsilon=16/255$) while maintaining much higher standard accuracy (an absolute advantage of over 9% than Fast-AT-GA in SA). Moreover, as shown in Table 4, our proposed method achieves the best robustness across different model architectures. Those results demonstrated the superiority of Fast-BAT in achieving SOTA robustness, mitigating catastrophic overfitting, and holding extraordinary stability. Besides the empirical results, our novelty also lies in the theoretical side; see our clarification in [General Response 1](https://openreview.net/forum?id=gzeruP-0J29&noteId=dZTezf6hoeP).
> > >
> > > [Assuming that all else is held equal (i.e. that the authors do not perform any new systems optimizations that previous work did not)]
> > > However, we do not believe that this assumption can be applied to our work. As clarified in [Highlighted General Response](https://openreview.net/forum?id=gzeruP-0J29&noteId=y4L0EA0nnP), Fast-BAT based on the bi-level optimization algorithm and implicit gradient yields a brand new optimization system for adversarial training. To the best of our knowledge, our paper takes a solid step in this direction for the first time. This is also an important theoretical contribution, which has been acknowledged by [Revewer TZX8](https://openreview.net/forum?id=gzeruP-0J29&noteId=6pEA1oN1eeM) and [Reviewer N6su](https://openreview.net/forum?id=gzeruP-0J29&noteId=a3laIqJRpdr).
> > >
> > > We sincerely hope Reviewer 6Qs9 could take both our empirical and theoretical contributions into full consideration. If you have any further questions, we are happy to discuss them with you.

---

### Author Response · Authors · 2021-11-19
**Highlighted general response**

Thank you very much for the very insightful comments. Please see the highlighted revisions and response below.


**Clarification on our contributions** (@Reviewer 6Qs9, @Reviewer c58a)

We further elaborate on our contributions below.

   1) It is our contribution to rigorously derive and interpret fast adversarial training (Fast-AT) through the lens of bi-level optimization (BLO) (Sec. 3). Note that in the past, Fast-AT was only studied in a heuristics-based fashion. To our knowledge, this is the first time that a framework is developed to offer a theoretical understanding of the Fast-AT algorithm.
Although the math we used is relatively simple, it is **nontrivial** to identify the relationship between the gradient sign-based PGD attack generation and the lower-level linearization of BLO.

   2) The proposed BLO view is useful since it allows the (lower-level) attack objective to be different from the (upper-level) training objective. This is the key to analyzing the Fast-AT-type methods from the perspective of lower-level linearization (Sec. 3 and 4).

   3) Analyzing the BLO-oriented Fast-BAT method is nontrivial since such a problem involves a constrained lower-level optimization problem. With this kind of constraint, it is no longer easy to compute the implicit gradient (Eq. 3) of the upper-level problem. By exploiting the problem structure of adversarial training, we leveraged the KKT conditions to acquire the closed-form expression of the implicit gradient in Theorem 1. Even in the line of optimization research, the work on how to efficiently solve lower-level constrained BLO problems is very limited. Thus, our optimization approach is novel. (Sec. 4)

   4) Experiment-wise, in addition to speeding up adversarial training, we have made a thorough experimental study by comparing our method with baselines in various adversarial learning setups, e.g., different victim models (Table 4), catastrophic robust overfitting vs. early stopping (Fig. 1), and Fast-BAT vs. gradient alignment (Fig. 2), Fast-BAT vs. gradient obfuscation (Table. 5). These provide a complete sanity check for the effectiveness of our proposal.

**Clarification on some possible misunderstandings**

1. [Reviewer 6Qs9] We have made detailed responses to justify the correctness and the fairness of our evaluation methods, which are also consistent with all baselines.

2. [Reviewer TZX8, Reviewer c58a] We further clarified and provided proofs for a number of our technical statements that may have caused some confusion. In our original submission, we intended to avoid some math-heavy material for improving readability. However, this may hurt preciseness. Reviewers' comments inspired us to further revise our technical statements and make them as precise as possible.

**Summary of newly conducted experiments**
Reviewers' insightful comments have inspired us to conduct a series of new experiments as summarized below.

1. [Reviewer 6Qs9] Following your suggestion, we showed the improvement of our approach over baselines in terms of maximum accuracy versus the total training time.

2. [Reviewer TZX8, Reviewer c58a] Following your suggestion, we compared the Hessian-free Fast BAT with its Hessian-involved version. We did not see much performance difference in the presence of Hessian.

3. [Reviewer N6su] We added the ablation study on $\alpha_2$. And we integrated BLO with the standard adversarial training and showed the comparison. Further, we evaluated the performance of our approach in neural networks using different types of activation functions.

---

### Author Response · Authors · 2021-11-22
**Summary of responses to all reviewers**

Dear Reviewers and Area Chairs,

Thank you very much for the efficient handling of our manuscript. In the rebuttal phase, we have made a substantial effort in providing additional experiment results and clarifications requested by reviewers. We hope that reviewers find the response valuable in addressing the raised questions. Below is a summary of our responses.

Reviewer [6Qs9](https://openreview.net/forum?id=gzeruP-0J29&noteId=qTX7OkYLwp):
1.  We clarified some possible mis-understandings about our contributions.
2.  We clarified our evaluation metrics that align with the literature.
3.  We presented new experimental results on the maximum standard accuracy (SA) and robust accuracy (RA) versus the amount of training time by treating the epoch number as a free parameter, following the reviewer's suggestion.
4.  We clarified 'lack of stability' and 'robust overfitting', the known issues of Fast AT, and provided some more background information on the terminologies used in the adversarial ML community.

Reviewer [TZX8](https://openreview.net/forum?id=gzeruP-0J29&noteId=6pEA1oN1eeM):
1.  We provided rigorous explanations associated with our statements in Sec. 3 and Sec.4.
2.  We conducted 5 more repetitive trials for the mean and variance values in Table 1.
3.  We clarified the poor accuracy-robustness tradeoff of Fast-AT-GA.
4.  We made rigorous mathematical analysis on our argument on the relationship between Problem (1) and Problem (2) when $\ell_{\mathrm{atk}}=-\ell_{\mathrm{tr}}$.
5.  We clarified the significance of our contribution to interpreting Fast-AT from the BLO perspective.
6.  We provided rigorous proof of Eq. (6).
7.  We clarified our Hessian-free assumption and conducted new experiments to empirically justify it.
8.  We clarified our statement on the case where the model parameter updating step of Fast-BAT reduces to Fast-AT when \alpha_2=0.

Reviewer [c58a](https://openreview.net/forum?id=gzeruP-0J29&noteId=DT9fJ9Ab8b5):
1.  We clarified our contribution to combining adversarial training with BLO and to solving the bi-level optimization problem with the lower-level constraints.
2.  We clarified some possible misunderstandings on the smoothness of projection function, the validity of chain rule as well as our derivation of implicit gradient based on KKT conditions.
3.  We explained the role of the strongly convex regularization term.
4.  We justified our Hessian-free assumption through new experiments.
5.  We provided more background information on catastrophic overfitting, gradient alignment, Fast-AT-GA, and PGD-2-AT.
6.  We conducted a significance test to justify the significance of our achieved improvement.

Reviewer [N6su](https://openreview.net/forum?id=gzeruP-0J29&noteId=a3laIqJRpdr):
1.  We clarified the role of $\alpha_2$ in trading off the robustness and accuracy of Fast-BAT.
2.  We explained the reason for a well-aligned gradient brought by Fast-BAT.
3.  We conducted new experiments to explore the possibility of implementing BLO on solving the standard adversarial training problems.
4.  We conducted new experiments to further justify the validity of Hessian-free assumption on non-ReLU neural networks.

---

### Decision · Program_Chairs · 2022-01-20

**Decision:**

Reject

**Comment:**

This paper investigates fast adversarial training methods as a bilevel optimization problem. The proposed algorithm compares well with the existing techniques in overall runtime (obtaining better clean-test accuracy, which is not the goal, and) matching the robust accuracy of existing adversarial training methods. The proposed framework, however, is more general and flexible and is theoretically grounded. The problem studied here is exciting and the approach the authors take is interesting.

The current version, unfortunately, has some serious shortcomings. The empirical comparisons are a bit lacking — in general, the wall clock time is not a very good measure, it depends heavily on the implementation and various optimizations therein. A more suitable comparison would be in terms of floating-point operations, or in terms of iteration complexity.

The paper reports other interesting findings such as how the proposed method avoids robust overfitting. However, there is little theoretical evidence or insight for how the proposed method avoids it.

The writing can be improved with more emphasis on the novelty and significance of the contributions — some of the statements regarding improvements over prior work are somewhat misleading given the incremental gains (e.g., see Table 1). I believe the comments from the reviewers have already helped improve the quality of the paper. I encourage the authors to further incorporate the feedback and work towards a stronger submission.